# Robust and consistent measures of pattern separation based on information theory and demonstrated in the dentate gyrus

**Alexander D. Bird** [1,2,3,4]*, **Hermann Cuntz**[2,3,4], **Peter Jedlicka**[1,4]

**1** Computer-Based Modelling in the field of 3R Animal Protection, ICAR3R, Faculty of Medicine, Justus Liebig University, Giessen, Germany, **2** Ernst Strüngmann Institute (ESI) for Neuroscience in cooperation with the Max Planck Society, Frankfurt-am-Main, Germany, **3** Frankfurt Institute for Advanced Studies, Frankfurt-am-Main, Germany, **4** Translational Neuroscience Network Giessen, Germany

* alex.neurosci@gmail.com

**Data Availability Statement:** Code for all figures (except Fig 5) is written in Matlab, code for Fig 5 is written in Python. Code is freely available for download (see S1 Code). The functions necessary

## Abstract

Pattern separation is a valuable computational function performed by neuronal circuits, such as the dentate gyrus, where dissimilarity between inputs is increased, reducing noise and increasing the storage capacity of downstream networks. Pattern separation is studied from both *in vivo* experimental and computational perspectives and, a number of different measures (such as orthogonalisation, decorrelation, or spike train distance) have been applied to quantify the process of pattern separation. However, these are known to give conclusions that can differ qualitatively depending on the choice of measure and the parameters used to calculate it. We here demonstrate that arbitrarily increasing sparsity, a noticeable feature of dentate granule cell firing and one that is believed to be key to pattern separation, typically leads to improved classical measures for pattern separation even, inappropriately, up to the point where almost all information about the inputs is lost. Standard measures therefore both cannot differentiate between pattern separation and pattern destruction, and give results that may depend on arbitrary parameter choices. We propose that techniques from information theory, in particular mutual information, transfer entropy, and redundancy, should be applied to penalise the potential for lost information (often due to increased sparsity) that is neglected by existing measures. We compare five commonly-used measures of pattern separation with three novel techniques based on information theory, showing that the latter can be applied in a principled way and provide a robust and reliable measure for comparing the pattern separation performance of different neurons and networks. We demonstrate our new measures on detailed compartmental models of individual dentate granule cells and a dentate microcircuit, and show how structural changes associated with epilepsy affect pattern separation performance. We also demonstrate how our measures of pattern separation can predict pattern completion accuracy. Overall, our measures solve a widely acknowledged problem in assessing the pattern separation of neural circuits such as the dentate gyrus, as well as the cerebellum and mushroom body. Finally we provide a publicly available toolbox allowing for easy analysis of pattern separation in spike train ensembles.

to assess pattern separation using both classical and information theoretic measures are included separately as the pattern separation toolbox on Github. In addition, raw optimal codes and binsizes are being uploaded to the Zenodo data repository (currently doi:10.5281/zenodo.10233025).].

**Funding:** This work was supported by the Bundesministerium für Bildung und Forschung (BMBF No. 579 031L0229 to PJ) and by the Deutsche Forschungsgemeinschaft (DFG No. 467764793, JE 528/10-1 to PJ). The funders had no role in study design, data collection and analysis, decision to publish, or preparation of the manuscript.

**Competing interests:** The authors have declared that no competing interests exist.

## Author summary

The hippocampus is a region of the brain strongly associated with spatial navigation and encoding of episodic memories. To perform these functions effectively it makes use of circuits that perform *pattern separation*, where redundant structure is removed from neural representations leaving only the most salient information. Pattern separation allows downstream pattern completion networks to better distinguish between similar situations. Pathological changes, caused by Alzheimer's, schizophrenia, or epilepsy, to the circuits that perform pattern separation are associated with reduced discriminative ability in both animal models and humans. Traditionally, pattern separation has been described alongside the complementary process of pattern completion, but more recent studies have focussed on the detailed neuronal and circuit features that contribute to pattern separation alone. We here show that traditional measures of pattern separation are inappropriate in this case, as they do not give consistent conclusions when parameters are changed and can confound pattern separation with the loss of important information. We show that directly accounting for the information throughput of a pattern separation circuit can provide new measures of pattern separation that are robust and consistent, and allow for nuanced analysis of the structure-function relationship of such circuits and how this may be perturbed by pathology.

## Introduction

The hippocampus plays an important role in a number of crucial behavioural functions [1], in particular encoding spatial information [2, 3] and episodic memories [4, 5]. To support this functionality, a portion of the inputs to the hippocampus are first processed in the dentate gyrus, a subfield consisting of large numbers of sparse-firing principal neurons, the granule cells, arranged in a consistent laminar structure [6, 7]. The dentate gyrus is one of the few brain regions to undergo adult neurogenesis [8–10], with adult-born cells displaying distinct intrinsic, synaptic, and morphological properties [11, 12]. A number of functional roles have been postulated for the dentate gyrus, but the most consistently studied and experimentally supported is that of pattern separation: an increase in the dissimilarity of outputs compared to inputs [13, 14].

### Pattern separation

The idea of pattern separation as a desirable computational function arises from the work of David Marr on the encoding of memories [15, 16]. Marr showed that a strongly recurrent neuronal network could learn to reproduce complete patterns from fragmented inputs, but that such fragments should be distinctive to avoid catastrophic interference between different patterns. Although the theory was developed for the cerebellum, another site of pattern completion was proposed to be the CA3 hippocampal subfield, a region that displays the necessary recurrent connectivity and receives strong inputs directly from the dentate gyrus [17, 18]. Further theoretical work, in particular that of McNaughton and Morris [19] and Rolls [20], showed that pattern completion would ideally be preceded by pattern separation, originally envisioned as an orthogonalisation of input spiking vectors, and that this could take place in the dentate gyrus. The theory of pattern separation, and its occurrence in the dentate gyrus, has since been extensively developed in a large number of theoretical papers [21–26] and observed in many experiments [27–31] (for reviews see [32, 14], and [33]). Pattern separation

has been viewed both in terms of the spiking outputs of the dentate gyrus and the ability of an experimental subject to discriminate between similar situations [34, 35]. Of particular clinical relevance, the dentate gyrus undergoes relatively well-described physiological changes under pathologies such as Alzheimer's disease [36], schizophrenia [37], and epilepsy [38–40], and these conditions have direct behavioural correlates in terms of the reduced ability of sufferers to distinguish between similar situations [41–43].

A number of measures of spike train similarity have been used to assess pattern separation in the literature. The original definitions of patterns separation considered orthogonalisation as most beneficial in terms of reducing interference during pattern completion [19, 20]; this is computed as the cosine distance between (normalised) discretised spike-train vectors. Decorrelation has also been used as a natural measure of similarity reduction [44, 45] and again is computed on discretised spike-train vectors. Finally, different measures of distances between spike trains can be used to assess pattern separation; the measures can be either discretised, such as the Hamming distance [46–49], or calculated on raw spike times [50–53].

Despite being conceptually straightforward, the quantification of pattern separation in terms of a network's inputs and outputs can be surprisingly problematic. When considered as part of a system preceding pattern completion, good pattern separation can be simply taken as any procedure that reduces noise and enhances the storage capacity of the downstream network [54]. When isolated from pattern completion, however, measurement of pattern separation efficacy can become more arbitrary. Different research questions, theoretical models, and neural codes can lead to wildly different conclusions about pattern separation. This has become particularly relevant as more detailed computational models have focussed on the dentate gyrus in order to assess the contribution of different circuit components to pattern separation [45, 46, 48, 49], and as new experimental techniques have allowed direct access to the inputs and outputs of individual dentate granule cells [44, 55]. A contributing factor to the number of different metrics is the gap between complex *in vivo* physiological studies which focus on the mechanisms generating pattern separation [56, 57] and more computational studies that focus on the consequences of pattern separation [58]. The former tend to dominate the literature. Different quantifications of the effect called pattern separation can lead to seemingly inconsistent conclusions, a point already noted by Santoro [34], Vineyard et al [59], and Chavlis and Poirazi [33]. Madar et al [52] showed explicitly that different measures can produce different results on the same datasets. Myers and Scharfman [46] found that removing hilar cells led to less sparse granule cell activity and improved pattern separation performance, as measured with the raw Hamming distance, a counterintuitive result given the theoretically positive effects of sparsity on pattern separation [60, 61]. Later studies that used the Hamming distance sought to correct for the relative sparsity of the outputs compared to the inputs in various ways [47, 49]. Sparsity in general can be a problem for similarity metrics as firing rates can bias correlations and cosine distances [51, 52, 62]. A particular issue arises in the choice of bin sizes to move from spike times to the discretised spike-vectors necessary to compute orthogonalisations, decorrelations, or Hamming distances; Madar et al [52] also showed that choosing different bin sizes could qualitatively change the apparent behaviour of a system, with smaller time bins typically showing pattern separation and larger time bins showing pattern convergence on the same input and output sets of spike times.

Finally, making structured inputs less similar can be achieved by destroying structure and this may degrade the ability of downstream networks, such as the CA3 subfield, to reconstruct salient features from the inputs. Vineyard et al [59] and Madar et al [44] both note that standard pattern separation measures do not consider destruction of information. This has lead to debates about the roles of different features of the dentate gyrus in pattern separation. Aimone et al [63] and Sahay et al [64] took explicitly opposing views on the function of neurogenesis in

the dentate gyrus, with the former arguing that 'memory resolution', the transfer of information salient to memories, was the key role of more broadly-tuned adult-born neurons and the latter arguing that more excitable adult-born neurons primarily drove recurrent inhibition that increased sparsity and hence classical pattern separation. The two objectives, reliable transmission and separation through sparsity, were considered to be largely antithetical. Inconsistencies in the conception and measurement of pattern separation contribute to the continuation of this debate; later arguments have been made that adult-born granule cells aid pattern separation [59, 65, 66], hinder pattern separation [67], differentially separate the most similar patterns [68], or independently carry out the entirety of pattern separation, with mature cells responsible for behavioural pattern completion in behavioural tasks [69] (but see [70] for criticism of this idea). Some studies consider the loss of signal explicitly. Guzman et al [45], for example, trained an artificial neural network to identify input patterns from modelled dentate gyrus outputs, whilst quantifying pattern separation itself by decorrelation until the stage that inputs cannot be reliably identified by the artificial network. This is an effective solution to the problem of complete loss of signal, but leaves open questions about how close the network is to failing, or, conversely, if a more complex artificial network could recover more input patterns for a given level of decorrelation. We believe that a direct measurement of the information contained in spike train ensembles going into and out of the dentate gyrus will help to resolve these inconsistencies and be a valuable and robust assessor of dentate gyrus functionality.

## Information theory in neuroscience

Information theory is built upon the work of Shannon [71] and studies the ability of systems to communicate reliably. There is a long history of the application of information theoretic techniques to neuroscience, starting with MacKayand McCulloch [72], and a large number of techniques have been developed to assess or estimate the informational content of neural codes [73–76] (see [77] or [78], among many others, for reviews). Since information has a natural link to the functions of the hippocampus in terms of memory, navigation, and situational discrimination, a number of studies have used techniques from information theory to analyse hippocampal structure and function. Treves and Rolls [79] showed that the CA3 network should receive two streams of input (from the dentate gyrus and directly from the upstream entorhinal cortex) in order to both learn new patterns and reliably access stored memories. Cerasti and Treves [80] found that the spatial representations induced in CA3 by dentate gyrus activation potentially contain much information that is so high-dimensional that it cannot be feasibly decoded. Petrantonakis and Poirazi [81] applied compressed sensing theory to show that the firing activity of different hippocampal regions was consistent with taking random projections to lower the dimensionality of a spike code and allow practical decoding performance substantially above the deterministic limit. Fagihi and Moustafa [48] applied mutual information as a measure to tune parameters of single neurons in a network model of schizophrenia, but assessed the resulting pattern separation functionality of the model using a form of Hamming distance. Vineyard et al [59] introduced a variety of loss-less coding strategies to estimate the maximal amount of information that could be encoded by a model dentate gyrus, and how this is increased by neurogenesis, but treated information maximisation as the ultimate goal rather than pattern separation as typically described. Severa et al [82] expanded on these ideas by describing a loss-less spike code for binary vectors that would increase sparsity and reduce correlations through an abstracted dentate-gyrus layer. In the cerebellum, Billings et al [58] studied how connectivity and cellular morphology support information throughput despite increased sparsity, concluding that these two features should be balanced. No previous

study has, however, applied information as a direct measure of pattern separation efficacy in the dentate gyrus for arbitrary inputs.

We here introduce a set of techniques from information theory that allow pattern separation in the dentate gyrus to be rigorously quantified without the need to specify an accompanying pattern completion algorithm. We first confirm that arbitrarily removing structure from input spike train ensembles typically improves pattern separation as assessed by classical measures, but reduces the information content. We then show that mutual information [71], transfer entropy [83], and redundancy [84] can be used as measures of pattern separation performance and apply these to detailed compartmental models of individual granule cells and a small network. We further demonstrate how information estimation techniques [76, 85] can be adapted to spike train ensembles in order to reduce biases when data is scarce [86]. We demonstrate how our new measures might better predict pattern completion accuracy in an abstract auto-associative model. Finally, we demonstrate how structural changes of the dentate circuit associated with epilepsy [38, 40] affect pattern separation performance as assessed by our new measures and how they might apply to a larger network. We anticipate that information-theoretic measures of pattern separation will be a useful tool in disentangling the structure-function relationships of different cell types, how these relationships arise from learning and development, and how they are perturbed by pathology.

## Results

### Sparsity typically increases standard measures of pattern separation, but decreases information transmission

Classical measures of pattern separation, such as orthogonalisation, decorrelation, or spike train distance, are effective at analysing the measured behaviour of a neuronal system and quantify the degree of difference between the inputs and outputs. In the dentate gyrus, outputs are typically encoded by sparser activity than the inputs and sparsity is a major contributor to pattern separation [21, 43, 44, 60, 82, 87, 88]. Increasing sparsity in a neural code, however, may reduce the information it carries about the afferent signal [58]. We here introduce a set of simple filters that arbitrarily increase the sparsity of an ensemble of spike trains and show both that classical measures of pattern separation can be highly parameter-dependent and that an arbitrary reduction in spike number typically leads to an increase in measured pattern separation performance.

Fig 1A (left) shows a raster of spike trains phase-locked to a periodic stimulus, and so displaying correlations both within and between spike trains (see Methods). Activity patterns consisting of ensembles of spike trains with different correlations within and between different trains are used throughout this study. Sparse trains are produced using thinning filters that remove spikes from the inputs in some way. Shown here is the independent uniform removal of individual spikes with a fixed probability $p$ (Fig 1A, right), but we also consider three more structured filters: *n-th pass*—only every $n$-th spike in each train can pass, *refractory*—delete every spike within $t$ seconds of a previous spike in the same train, and *competitive*—delete every spike within $t$ seconds of another spike in *any* train in the ensemble (S1(B) Fig). These different filters pick out different relationships in the input spike train ensemble. The first filter destroys all structure in the ensemble as the deletion probability increases, the second and third filters maintain order or temporal structure within trains, and the fourth corresponds to competition between different trains in the ensemble.

We next apply five metrics that are regularly used to assess pattern separation in the literature to spike train ensembles filtered in these ways. Overall this study compares three novel information-based and five existing measures of pattern separation. In all cases we use the

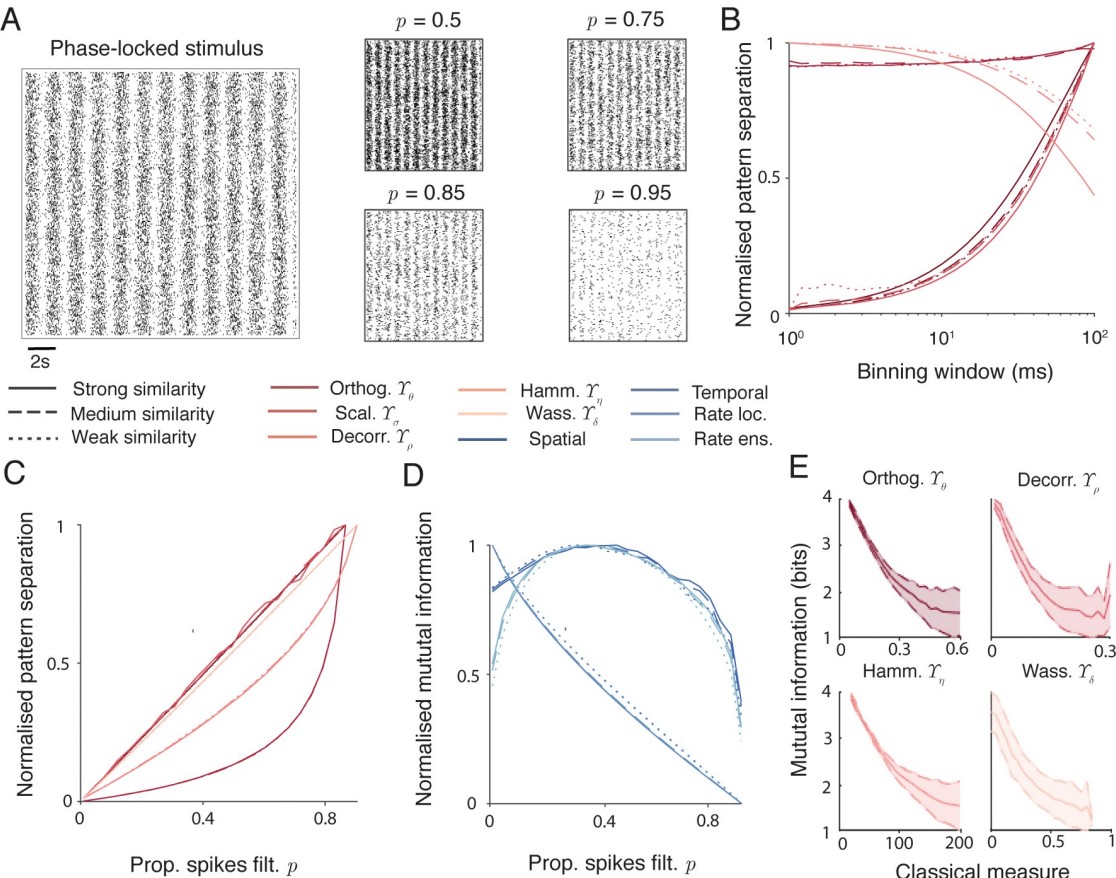

**Fig 1. Sparsifying (filtering) spike train ensembles leads to increased classical measures of pattern separation, but reduced information content. A** Demonstration of filtering a phase-locked spike train ensemble (left) by random spike deletion (right). The probabilities of spike deletion are $p$ = 0.5, 0.75, 0.85, 0.95. **B** Standard measures of pattern separation as a function of discretisation bin size on a randomly filtered spike train with $p$ = 0.5. Solid, dashed, and dotted lines refer respectively to strong, medium, and weak input similarities (phase-locked strengths of 0.75, 0.5, and 0.25, see Methods). From darkest to lightest, colours plot standard measures of pattern separation: orthogonalisation $\Upsilon_\theta$, scaling $\Upsilon_\sigma$, decorrelation $\Upsilon_\rho$, and Hamming distance $\Upsilon_\eta$. Wasserstein distance $\Upsilon_\delta$ does not rely on discretisation and is not plotted. All values are normalised for comparisons, raw values are plotted in S1(C)–S1 (F) Fig. **C** Standard measures of pattern separation applied to filtered spike trains. Filtering methods and input similarities are as in **B**. The $x$-axis gives the filtering parameter in each case. All values are normalised for comparisons, raw values are plotted in S1(G)–S1 (K) Fig. **D** Mutual information between input spike train ensembles and filtered spike train ensembles. Filtering methods and input similarities are as in **B**. Colours correspond to different neuronal codes (see Methods): from darkest to lightest instantaneous spatial, temporal, local rate, and ensemble rate. All values are normalised for comparisons, raw values are plotted in S2(A)–S2(D) Fig. **E** Mutual information as a function of classical pattern separation measures. Mutual information was maximised over all spiking codes and bin sizes for different input rates, correlation structures and strengths, and filter types and strengths. Clockwise from top left: orthogonalisation $\Upsilon_\theta$, decorrelation $\Upsilon_\rho$, Wasserstein distance $\Upsilon_\delta$, and Hamming distance $\Upsilon_\eta$. Shaded areas show one standard deviation. Raw scatter plots and densities are shown in S2(E)–S2(H) Fig.

letter $\Upsilon$ to refer to a measure of pattern separation; upper case latin subscripts denote the information-based measures and lower case greek subscripts denote classical non-information based measures. The classical measures are plotted in progressively lighter shades of red in Fig 1B. Orthogonalisation $\Upsilon_\theta$ measures the reduction in pairwise cosine distance between normalised spike trains (Eqs 4 & 5), scaling $\Upsilon_\sigma$ measures the change in differences in the norms of the spike trains (Eqs 6 & 7), decorrelation $\Upsilon_\rho$ measures the reduction in pairwise correlations between spike trains (Eqs 8 & 9), Hamming distance $\Upsilon_\eta$ measures the increase in pairwise Hamming distance between spike train vectors (Eqs 10 & 11), and Wasserstein distance $\Upsilon_\delta$ measures the increase in pairwise Wasserstein distance between spike trains (Eqs 12 & 13).

**Table 1. Table summarising classical pattern separation measures (top), and new information theoretic measures (bottom).**

| Symbol | Definition | Description | Equations |
|---|---|---|---|
| $\Upsilon_\theta$ | Orthogonalisation | Cosine distance between normalised discretised spike train vectors | 4 & 5 |
| $\Upsilon_\sigma$ | Scaling | Ratio between norms of discretised spike train vectors | 6 & 7 |
| $\Upsilon_\rho$ | Decorrelation | Pearson correlation between discretised spike train vectors | 8 & 9 |
| $\Upsilon_\eta$ | Hamming distance | Hamming distance between discretised spike train vectors | 10 & 11 |
| $\Upsilon_\delta$ | Wasserstein distance | Wasserstein distance between raw spike time vectors | 12 & 13 |
| $\Upsilon_M$ | Mutual information | Sparsity weighted MI between input and output ensembles | 1 & 14 |
| $\Upsilon_T$ | Transfer entropy | Sparsity weighted TE between input and output ensembles | 2 & 15 |
| $\Upsilon_R$ | Redundancy reduction | Relative redundancy reduction between input and output ensembles | 3 & 16 |
| $\tilde{\Upsilon}_M$ | Estimated mutual information | Estimated $\Upsilon_M$ using a modified Kozachenko-Leonenko estimator | 1 & 17 to 19 |
| $\tilde{\Upsilon}_R$ | Estimated redundancy | Estimated $\Upsilon_R$ using a modified Kozachenko-Leonenko estimator | 3 & 17 to 19 |

These measures are fully described in Methods and summarised in Table 1. Orthogonalisation $\Upsilon_\theta$ and scaling $\Upsilon_\sigma$ are complementary measures as orthogonalisation is computed on normalised vectors and scaling compares norms. The first four ($\Upsilon_\theta$, $\Upsilon_\sigma$, $\Upsilon_\rho$, and $\Upsilon_\eta$) are computed on discretised spike trains, whereas the Wasserstein distance $\Upsilon_\delta$ is parameter-free and computed on raw spike times. There are other ways to apply such metrics to pattern separation. The Hamming distance, for example, can also be applied to differences between entire ensemble patterns rather than individual spike trains in an ensemble [46, 49]; we take this form to keep the measures used as consistent as possible.

Fig 1B plots the value of each measure (except the Wasserstein distance) normalised to its highest value (S1 Fig plots the unnormalised values in separate panels) for a random thinning with $p = 0.5$ as a function of the size of the discretisation bins used. In each case there is a monotonic relationship between the size of the bin and the value of the measure. This confirms the results of Madar et al [52] and means that it is very difficult to assign meaning to a given value of pattern separation computed by one of these methods as the value will typically depend on the choice of the binning parameter with no principled way to make this choice. It may be possible to pick approximate bin sizes based on biologically relevant timescales in some cases, potentially considering multiple different bins corresponding to multiple different timescales, but the precise width of the bins would remain a potential confound for any results.

Fig 1C plots the value of each measure normalised to its highest value (S1 Fig plots the unnormalised values in separate panels) as a function of spike deletion probability $p$. In all cases, each metric increases monotonically with the strength of the thinning filter and reaches a maximum when the output spike train ensembles are almost empty. This trend also holds for the other thinning filters (S1(G)–S1(K) Fig), with two exceptions: the Wasserstein distance measure $\Upsilon_\delta$ takes a maximum at an intermediate value of the refractory filter (S1(K) Fig), and the decorrelation measure $\Upsilon_\rho$ takes a maximum at an intermediate value of the competitive filter, where spike trains inhibit each other (S1(J) Fig). As the Wasserstein distance is rate-invariant (normalised to the total number of spikes in each train), it appears that removing spikes that lie close to other spikes can lead to lower measured distances [53]. For the decorrelation measure, the competitive filter removes spikes close to others in all parts of the ensemble and so can lead to very sparse vectors that are actually more correlated in the chosen time windows.

The potential problem with such measures in isolation is illustrated clearly in Fig 1D, which plots the normalised mutual information between input and output spike train ensembles (S2

Fig plots the unnormalised values in separate panels). Different possible spiking codes (instantaneous spatial, temporal, local rate, and ensemble rate, see Methods) are plotted in different shades of blue. Although many of these codes also depend on binning spikes, it is possible to pick bin sizes in a principled way to maximise the information content. This is done separately for each condition. Pattern separation using the above classical measures increases as information is lost. This is a general trend and holds for different input correlation structures and strengths, and mean firing rates. Fig 1E supports this point by plotting the maximum mutual information over all spiking codes and bin sizes as a function of different classical measures for general spike train ensembles and filters. There is a universal negative relationship, meaning that increased classical measures of pattern separation typically imply an increased loss of information about the input.

## Information theoretic measures of pattern separation

To avoid the above weaknesses of classical measures, we introduce two types of information theoretic measure of the ability of a neuronal system to separate patterns. The first type covers the relationship between inputs and outputs; giving a measure of the efficiency of the feedforward information transmission. The second type covers the relationships within the sets of inputs and outputs; giving a direct measure of the amount of information shared by an ensemble of spike trains which is more equivalent to the traditional definitions of pattern separation by orthogonalisation or decorrelation.

For the first type of measure, two related quantities are typically used to quantify the information throughput of a neural system: the symmetric mutual information $I_{X,Y}$ between input $X$ and output $Y$ ([71], Eq 14), and the directed transfer entropy $T_{X \to Y}$ from input $X$ to output $Y$ ([83], Eq 15). Both have their advantages and a recent review found that the transfer entropy is typically less biased when data is sparse [89].

The second type of measure is linked to a quantification of the redundancy within a spike-train ensemble. The principle comes from the partial information decomposition of an ensemble into information that is encoded synergistically, independently, and redundantly by different components of that ensemble [90–92]. Removing redundancy between spike trains would distinguish them in terms of the information they convey and improve pattern separation. Whilst multiple measures of redundancy exist, we follow Williams and Beer [84] in defining it as the minimum mutual information $R_X$ between different parts of a pattern $X$. Here this idea is applied as the minimum mutual information between each individual spike train and the rest of the ensemble (Eq 16).

The two types of informational measure can be used and combined to give robust measures of pattern separation. The information theoretic measures of pattern separation below are chosen to take a value of zero both in the limit of destroying all information between input and output, and when inputs have not been differentiated at all. To directly adapt the first type of measure to assess pattern separation, we take sparsity as a parameter-free proxy for pattern separation when combined with mutual information (Eqs 1 and 2). It would also be possible to take a traditional measure such as orthogonalisation or decorrelation and penalise this with information loss. The simple sparsity measure is preferred to these classical measures as it makes no assumptions on the pattern completion circuit, requires no discretisation of spike times, and has a natural relationship to the metabolic costs of spiking. The sparsity weighted measures therefore correspond to a more efficient coding of input patterns, likely using a higher-dimensional coding space [81, 93] where patterns will likely be more separated [58, 61, 88]. These measures can take negative values in the cases where the output contains more spikes than the input (ie the sparsity is 'negative'), and can take similar values in cases where

information throughput is high but sparsity is low and where information throughput is low but sparsity is high. There can also be value in considering the two components separately to see how far pattern separation performance is driven by maintenance of information throughput and how far by reduction in input similarity (see Results below).

To adapt the second type of information theoretic measure to assess pattern separation, we multiply the reduction in redundancy between the input ensemble and the output ensemble with the mutual information between the pair of ensembles, balancing feedforward information loss with reduced redundancy (Eq 3). All information-theoretic measures, as well as the classical measures discussed here, can be computed by our accompanying Matlab pattern separation toolbox. This toolbox combines both the calculation of the new metrics themselves, which depend on choices of neural code and numerical parameters such as discretisation binsize, and the optimisation algorithms we use to identify the codes and parameters that allow the maximum amount of information to be decoded.

**Sparsity weighted mutual information $\Upsilon_\mathbf{M}$.** If the number of spikes in the input pattern $X$ is $m_X$, the number in the output pattern $Y$ is $n_Y$, and $I_{X,Y}$ (Eq 14) is the mutual information between $X$ and $Y$, then the sparsity weighted mutual information is

$$\Upsilon_\mathrm{M} = \frac{m_X - n_Y}{m_X} I_{X,Y} \tag{1}$$

The reason for normalising the sparsity to the number of input spikes is to allow for the fact that sparse input encodings have relatively less room to remove spikes. We note again here that sparsity on its own is not a measure of pattern separation, but can be combined with mutual information $I_{X,Y}$ to produce a useful proxy.

**Sparsity weighted transfer entropy $\Upsilon_\mathbf{T}$.** Similarly, if $T_{X \to Y}$ is the transfer entropy from $X$ to $Y$, then the sparsity weighted transfer entropy is

$$\Upsilon_\mathrm{T} = \frac{m_X - n_Y}{m_X} T_{X \to Y} \tag{2}$$

**Relative redundancy reduction $\Upsilon_\mathbf{R}$.** Comparing the difference in redundancies between the input and output ensembles gives a measure of the changes taking place *within* a pattern as it passes through a neural system. This can be multiplied by the mutual information between input $X$ and output $Y$ to measure the balance between redundancy reduction and information throughput

$$\Upsilon_\mathrm{R} = (R_X - R_Y) I_{X,Y} \tag{3}$$

Redundancy reduction $R_X - R_Y$ alone can be seen as its own measure of pattern separation. It is more analogous to the the classical measures described above as it only describes the relationships within the input and output spike train ensembles separately. It measures the reduction in input similarity in terms of the informational quantity of redundancy, instead of the correlations or cosine or Wasserstein distances between spike trains in an ensemble. It also shares the key issue of neglecting information throughput from the input to the output, so we suggest that relative redundancy reduction $\Upsilon_R$ should be taken as a better measure of pattern separation performance.

**Information theoretic measures of pattern separation and filtered spike ensembles.** Fig 2 shows the application of these measures to filtered spike train ensembles as in Fig 1. Fig 2A shows the sparsity weighted mutual information $\Upsilon_\mathrm{M}$, Fig 2B the sparsity weighted transfer entropy $\Upsilon_\mathrm{T}$, and Fig 2C the relative redundancy reduction $\Upsilon_\mathrm{R}$. In all cases there is a single peak in each measure at an intermediate value of the thinning filters, and this also holds for the

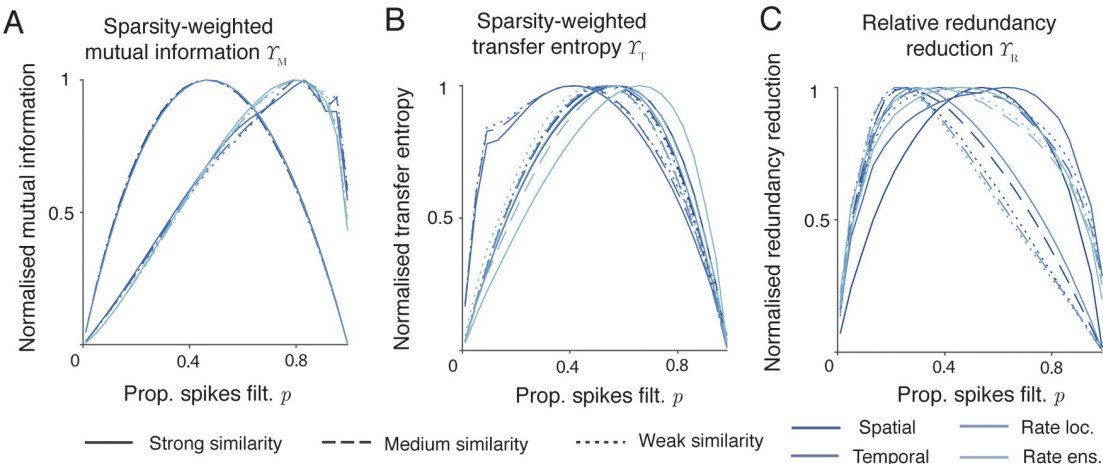

**Fig 2. Information theoretic measures of pattern separation penalise pattern destruction (information loss). A** Normalised sparsity weighted mutual information $\Upsilon_M$ applied to randomly filtered spike train ensembles. Solid, dashed, and dotted lines refer respectively to strong, medium, and weak input similarities (see Methods). Colours correspond to different neuronal codes (see Methods): from darkest to lightest instantaneous spatial, temporal, local rate, and ensemble rate. All values are normalised for comparisons, raw values are plotted in S3(A) Fig. **B** Normalised sparsity weighted transfer entropy $\Upsilon_T$ applied to filtered spike train ensembles. Colours as in panel **A**. All values are normalised for comparisons, raw values are plotted in S3(B) Fig. **C** Normalised relative redundancy reduction $\Upsilon_R$ applied to filtered spike train ensembles. Colours as in panel **A**. All values are normalised for comparisons, raw values are plotted in S3(C) Fig.

$n$th-pass, refractory, and competitive filters (S3 Fig). As before, different assumed spiking codes lead to different relationships between sparsity and information transmission (different shades of blue in panels A to C), but the principled approach is to take the codes that produce the highest information throughput. For $\Upsilon_M$ and $\Upsilon_T$, the measures are relatively independent of the strength of the phase-locking within the input patterns (see S3 Fig for unnormalised values). $\Upsilon_R$, however, which accounts for the reduction in redundancy between input and output patterns, is strongly dependent on the strength of input similarity (here the strength of the phase-locking), with stronger similarities leading to higher measures of pattern separation at all thinning filter values. Fig 2 plots the informational measures for a number of different spike codings, with parameters chosen to maximise each amount of information for each code. The optimal parameters, for both the inputs and outputs, differ for each datapoint, but the overall trend in informational terms is preserved. In general, we will also optimise over a number of different possible encodings; the goal is to identify the maximum amount of information that could possibly be contained in a given signal and so correctly penalise the absolute loss of information, however measured. This does not require an assumption that the spiking codes used by the entorhinal cortex maximise information efficiency, merely that it might be possible for a circuit to decode all of the information in a spike train ensemble.

$\Upsilon_M$ is a simple to understand and easy to apply proxy for pattern separation; it can be replaced by $\Upsilon_T$ if desired or if data is sparse (but see below for a procedure to estimate the mutual information in this case). $\Upsilon_R$ provides a useful complementary measure with more sensitivity to the strength and structure of input correlations.

## Single cell pattern separation

Whilst full pattern separation likely relies on circuit structure [5, 45], features of single cells embedded in their networks can contribute to the separation of patterns in time [44]. If a single neuron is sequentially presented with similar input patterns and maps these to less similar

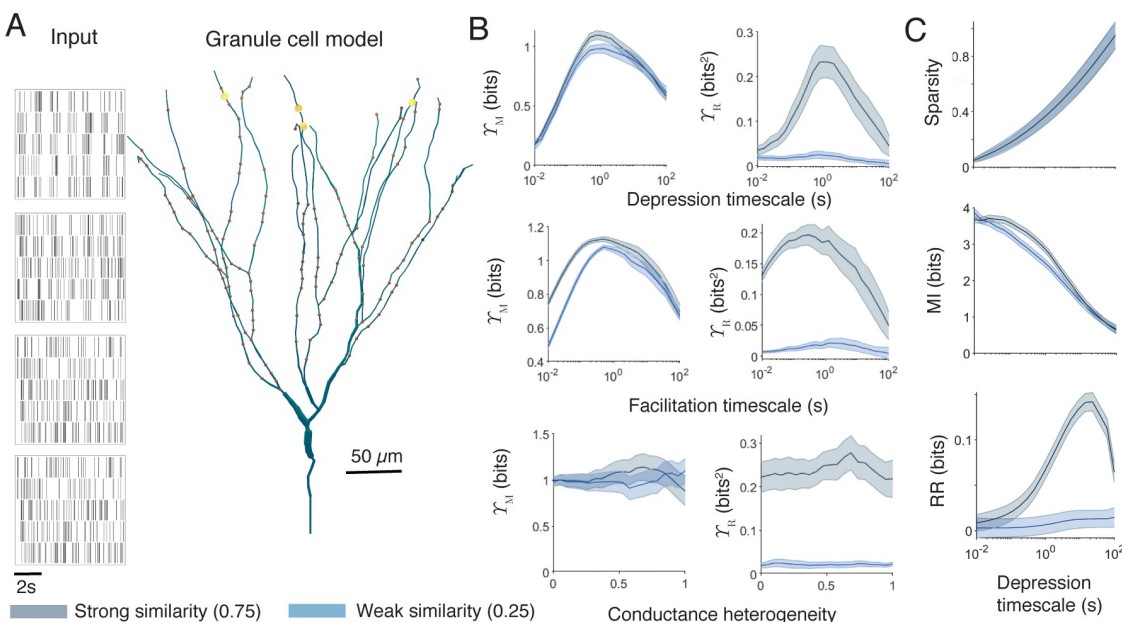

**Fig 3. Information theoretic measures applied to a single cell provide consistent and robust assessment of pattern separation. A** Left panels: Example input spiking rasters. Right panel: Dentate gyrus granule cell morphology. Informative synaptic contacts are shown by yellow and background synaptic contacts by brown markers. Example output voltage traces from the granule cell soma are shown in S4(A) Fig. **B**. Sparsity weighted mutual information $\Upsilon_M$ (left) and relative redundancy reduction $\Upsilon_R$ (right) for the granule cell model as a function of various physiological parameters. Spiking codes are chosen separately for the inputs and outputs to maximise information. From top to bottom: timescale of synaptic depression, timescale of synaptic facilitation, and spatial heterogeneity in ion channel densities (see Methods). Each input is presented 8 times. Solid lines represent the mean over 10 repetitions with different inputs and the shaded areas show one standard deviation above and below the mean. In general, input and output measures were close to the mean. Blue shows a weak input similarity (phase-locked correlation strength of 0.25), and grey a strong input similarity (phase-locked correlation strength of 0.75). Spike traces are two minutes long and consist of phase-locked inputs with a phase rate of 0.6Hz and a spiking rate of 5Hz. **C** Components of pattern separation as a function of the timescale of synaptic depression. From top to bottom: sparsity $(m_X - n_Y)/m_X$, mutual information $I_{X,Y}$, and redundancy reduction $R_X - R_Y$ (see Eqs 1 to 3).

output patterns, then measures of pattern separation can be computed on the combined input and output ensembles. This would be equivalent to a population of independent neurons simultaneously receiving different elements of the input ensemble and simultaneously producing different elements of the output ensemble.

To demonstrate the application of these new measures to simulated data from physiologically realistic single cell models, we study the effects of changing parameters on the ability of a detailed compartmental model of a single mouse granule cell [94] to separate patterns (Fig 3A). The cell receives informative synaptic input from the lateral perforant path (synaptic contacts shown as large yellow markers in Fig 3A), and background input across the rest of its dendritic tree (brown markers in Fig 3A). Inputs are presented as repeated structured patterns (rasters on the left of Fig 3A). Specifically, each spike train in the ensemble is presented sequentially and the output of the neuron is recorded. These outputs are combined into an output ensemble. Perforant path inputs to granule cells undergo stochastic short-term plasticity (see Methods for details), and varying the timescales of depression and facilitation alters both the sparsity-weighted mutual information $\Upsilon_M$ and relative redundancy reduction $\Upsilon_R$ (Fig 3B, top two panels). Both measures show a consistent and robust peak in pattern separation at timescales of $\sim$ 500ms for depression and $\sim$ 300ms for facilitation, indicating an optimal parameter value for separation of the input patterns. The experimentally estimated values

for these timescales at lateral perforant path synapses in Madar et al [44] are $\sim$ 500ms for depression and $\sim$ 9ms for facilitation. Similarly, the public Hippocampome.org dataset [95] gives average recorded timescales of 563.9ms for depression and 8.33ms for facilitation at the time of writing. The sparsity weighted mutual information measure $\Upsilon_M$ is relatively unaffected by changes in the similarities of the input patterns compared to the relative redundancy reduction measure $\Upsilon_R$, which shows higher values when input similarities are stronger. The peaks of the curves are not affected by the strength of the input similarity when changing the depression timescales, but stronger similarities seem to slightly favour slower facilitation, with a rightwards shift in both the $\Upsilon_M$ and $\Upsilon_R$ curves. Using classical measures of pattern separation (see S4 Fig) does not reveal a consistent peak in performance, with most measures being either flat or monotonic across the range of timescales and usually giving results that depend on the size of discretised bin used.

In contrast to synaptic properties, the pattern separation performance of the cell is relatively unaffected by increasing intrinsic heterogeneity. The bottom row of Fig 3B shows pattern separation performance as a function of spatial ion channel heterogeneity. This is implemented by randomising the densities of the model's ion channels around their mean in each $1\mu m$ section of the neuron with a given coefficient of variation (see Methods). The lines for both measures are relatively flat, with no substantial impact of spatial heterogeneity on either $\Upsilon_M$ or $\Upsilon_R$. Changing the spatial heterogeneity in this way does not, on average, affect the excitability of the cell, but does alter the relative impacts of different synaptic contacts. In this model this change is insufficient to significantly alter the pattern separation performance.

After assessing the overall pattern separation performance, it is possible to investigate how the different components of our new measures may contribute. Fig 3C plots how sparsity, pure mutual information, and pure redundancy reduction change as a function of the timescale of synaptic depression. Sparsity is a purely increasing function, mutual information is a purely decreasing function, and redundancy reduction itself takes a peak for relatively slow vesicle recovery. Although such an analysis is helpful in identifying how a system is performing different aspects of pattern separation, it is not typically possible to identify the parameters that might give peak performance as shown by the combined measures $\Upsilon_M$ and $\Upsilon_R$.

Synapses with stochasticity and short-term plasticity are believed to contribute to pattern separation [44], gain modulation [96, 97], and energetically efficient information throughput [98]. The new techniques introduced here show how altering the timescales of this plasticity quantitatively changes the pattern separation performance of a single granule cell, and how experimentally measured timescales appear to be highly effective for pattern separation by balancing reduction of redundancy with loss of mutual information.

### Estimators for mutual information and performance on sparse data

A limitation of some informational measures is that they can require large amounts of data to produce accurate estimates of information content [86, 89]. This issue has been addressed by the development of estimators for informational quantities based on some measure of the distances between spike trains. In the Methods section we describe how to adapt the modified Kozachenko-Leonenko estimator [85] described in Houghton [76] to account for spike train ensembles and measurement of redundancy (Eqs 17 to 19). To apply the estimate consistently we also sought to remove a key parameter of the original implementation, the number of segments a spike train is divided into. We found that this parameter was unnecessary by adapting the technique Strong et al [75] introduced to estimate the limiting mutual information of infinitely long spike trains. Fig 4A (left panel) shows that, for large numbers of segments (see Methods), the mutual information estimate is a linear function of the number of segments. By

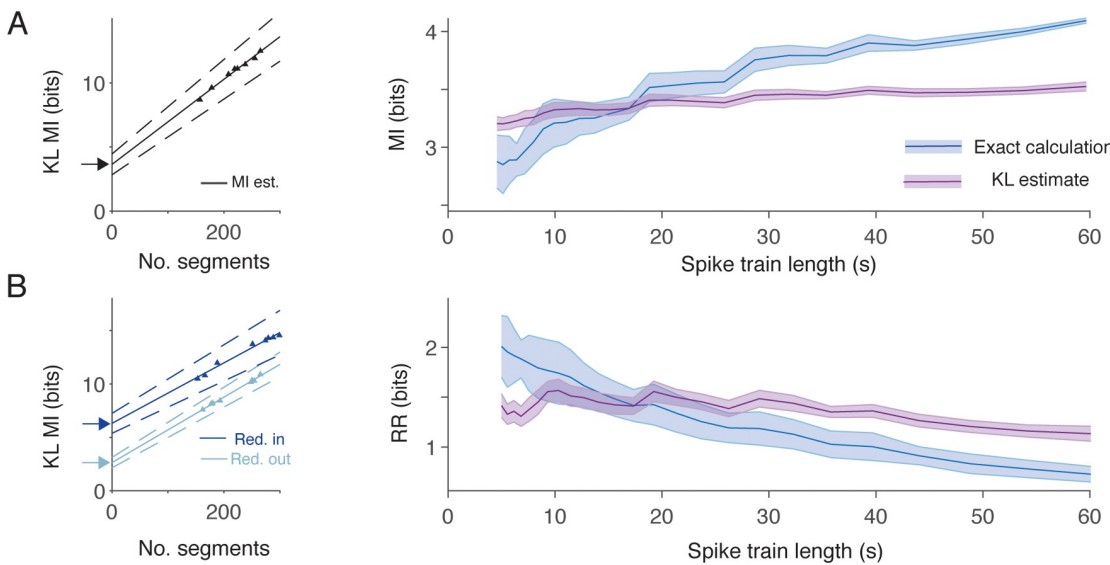

**Fig 4. Mutual information and redundancy can be estimated in situations of limited data. A** Left: Example point Kozachenko-Leonenko (KL) estimates (triangles) of mutual information (MI) for a spike train as a function of the number of segments (see Methods) and the extrapolation to infinite length (solid line and arrow). Dashed lines show the 95% confidence interval of the estimate. Right: Mean and standard deviation (shaded areas) in mutual information estimates for single-trial data as a function of spike train length. Blue shows the exact calculation using a temporal code and purple the Kozachenko-Leonenko estimate. **B** Left: Example point Kozachenko-Leonenko (KL) estimates (triangles) of redundancy (Red.) for a spike train as a function of the number of segments and the extrapolation to infinite length (solid lines and arrows). Dark blue shows the input ensemble (in) and light blue the output ensemble (out). Right: Mean and standard deviation (shaded areas) in redundancy reduction (RR) estimates for single-trial data as a function of spike train length. Colours as in panel **A**.

extrapolating back to the case of 0 segments, which must therefore be infinitely long, we find an estimate of the limiting mutual information of a spike train. A similar technique can be used for redundancy estimates (Fig 4B (left panel)).

By replacing the exact quantities in Eqs 1 and 3 with their estimators, this leads to estimators for sparsity weighted mutual information and relative redundancy reduction denoted $\tilde{\Upsilon}_M$ and $\tilde{\Upsilon}_R$ respectively. These estimators have advantages when data is sparse. Fig 4 (right panels) plots the distribution of estimates of mutual information and redundancy reduction using both the explicit temporal code (blue) and the Kozachenko-Leonenko estimators (purple) when only a single short trace over five spike trains is available. The standard deviation in mutual information estimates using the estimator is consistently smaller than the explicit calculation. Even when experimental data is sparse therefore, there are reliable ways to estimate pattern separation efficacy using our new information theoretic techniques. In specific circumstances, with relevant preliminary data or good estimates of circuit features such as input patterns and connectivity, it would be possible to use simulations such as this to constrain the length of the experiments needed to produce accurate exact values of $\Upsilon_M$ and $\Upsilon_R$.

The estimators $\tilde{\Upsilon}_M$ and $\tilde{\Upsilon}_R$ can be computed by our accompanying Matlab pattern separation toolbox.

## Information theoretic measures can predict pattern completion

A major strength of the information theoretic approach to measuring pattern separation is that it allows one to remain agnostic about the nature of any mechanisms of pattern completion in downstream circuits. Nevertheless, it is possible to demonstrate that our new measures

have value in quantifying the ability of an abstract pattern completion circuit to accurately recover inputs. The Hopfield network is a classical autoassociative network that is able to reconstruct complete patterns from partial inputs once it has been trained [99]. Hopfield networks suffer from catastrophic interference when learning to reproduce multiple patterns, and the extent of this interference typically depends on the degree of correlation between the trained patterns [100].

To evaluate how well information theoretic measures of pattern separation might predict pattern completion in this abstract model, we took the Kuzushiji-49 dataset of 49 handwritten Japanese Hiragana characters that is widely-used as a benchmark for computer vision algorithms [101]. The digits are encoded as 28 pixel by 28 pixel grayscale images, or as vectors of length 784. This means that the digit is represented by a population code. As handwritten digits, the elements of the dataset contain spatial correlations that impair the storage capacity of a Hopfield network. To train the network, the patterns are binarised by thresholding the grayscale values, and an exemplar of each class is chosen at random (S5(B) Fig). A Hopfield network with 784 nodes is then trained on the set of 49 exemplar patterns using the pseudo-inverse learning rule [102]. To assess the recall accuracy of the trained network, it is presented with each pattern in the full dataset and left to converge. The recalled class is then taken to be the class of the exemplar that best matches (has the greatest dot product with) the converged pattern. The initial recall accuracy is 16.83%, reflecting the problems caused by the correlated patterns.

Whilst it is possible to apply a basis transformation to the dataset to achieve high pattern completion performance, noise can also be used to separate the patterns. Noise is applied by randomising a subset of the pixels at fixed locations in each class (Fig 5A). This provides a noisy transformation on the input patterns; pattern separation metrics can be computed on this transformation (see Methods). Increasing the number of randomised pixels initially

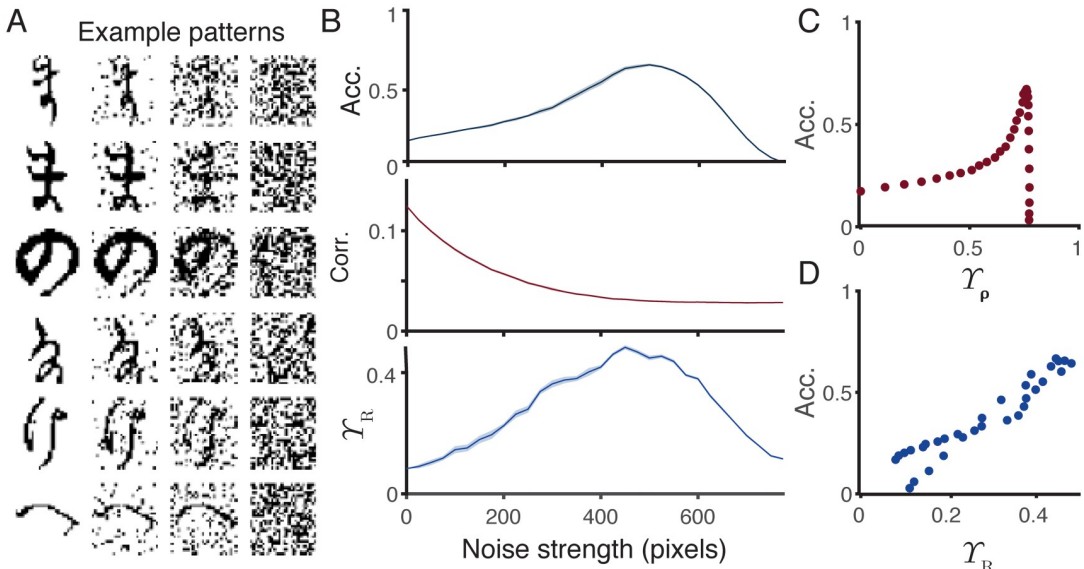

**Fig 5. Information theoretic measures can predict pattern completion performance. A** Example binary patterns from the Kuzushiji-49 dataset (top to bottom) with increasing levels of pixel noise (left to right: 0, 200, 400, and 600 pixels). **B** Top: Completion accuracy of a Hopfield network as a function of noise strength. Middle: Correlations between different classes of pattern as a function of noise strength. Bottom: Relative redundancy reduction ($\Upsilon_R$) between classes of patterns as a function of noise strength. Solid line shows the mean over 1000 repetitions and the shaded area shows the standard error. **C** Completion accuracy of a Hopfield network against decorrelation $\Upsilon_\rho$. **D** Accuracy against $\Upsilon_R$.

increases the performance of the network, whilst reducing correlations (Fig 5B). Beyond a certain point, however, the noise dominates and completion accuracy falls quickly towards a chance level. This is reflected in the relative redundancy reduction $\Upsilon_R$ of the noisy pattern transformation, which also takes a peak at an intermediate level of noise. As the patterns are not encoded by spikes, the sparsity-weighted mutual information $\Upsilon_M$ is not appropriate for this system. Directly comparing $\Upsilon_\rho$ and $\Upsilon_R$ with the pattern completion accuracy (Fig 5C and 5D) shows that, in this abstract model, $\Upsilon_R$ can be a useful predictor of subsequent pattern completion performance but $\Upsilon_\rho$ has poor predictive power.

This abstract model has a number of important qualitative differences from the real hippocampus. In particular, patterns are purely binary and presented instantaneously and separately to the network, with no background interference from other processes. Further, training and recall patterns pass through exactly the same transformation. In reality, hippocampal pattern completion circuits receive two streams of input, through the dentate gyrus and directly from the entorhinal cortex. These two streams likely have different roles in training and recall [1, 5, 79]. In addition, the model only covers one type of (population) code, and one type of pattern completion circuit. Nevertheless, the model illustrates how reducing redundancy whilst maintaining mutual information can be beneficial for storage and recall in a recurrent network and provides further motivation for our new metrics.

## Pattern separation in a network

Most studies of the dentate gyrus find that effective pattern separation in the dentate gyrus is primarily a network phenomenon [5, 61] produced by effects such as unreliable expansion to multiple principal cells, conditional reinforcement through lateral excitation, and competitive inhibition by interneurons [45, 61]. In addition, heterogeneities in the response properties of individual granule cells are likely to contribute [103]. We next show the applicability of our information theoretic pattern separation measures to the output of a simple microcircuit model of the mouse dentate gyrus. The full details are given in Methods, but overall we consider a population of 40 mature granule cells, 10 adult-born granule cells, 5 mossy cells, and one pyramidal basket interneuron (Fig 6A and 6B). The principal cells receive informative inputs in the outer molecular layer undergoing stochastic short-term plasticity as above, 5 groups are formed by sets of 8 mature and 2 adult-born cells receiving the same informative presynaptic spikes, but with independent stochastic synaptic dynamics. The output is taken to be the ensemble of all granule cell spike trains.

We first apply our measures to evaluate the relative pattern separation performance of mature and adult-born granule cells in the model. Fig 6C plots the sparsity weighted mutual information $\Upsilon_M$ and relative redundancy reduction $\Upsilon_R$ for both populations as a function of the strength of input similarities. All informational quantities are maximised over the relevant numerical parameters and the set of possible spike codes we consider. By both measures the pattern separation efficacy of the mature population is higher than that of the adult-born population in absolute terms, with a similar trend across both populations and measures in response to increasing input similarities. The increase in $\Upsilon_R$ with input similarity in the mature population is due to the fact that higher levels of input similarity imply higher absolute levels of input redundancy. The adult-born cells do not display any significant reduction in input redundancies at any similarity level. The reason for this appears to be that the adult-born cells in this model are more intrinsically excitable than the mature cells [94], and the modelled perforant path inputs are not scaled down to match this. There is evidence that excitatory inputs to adult-born cells in the real dentate gyrus are weaker to balance their higher intrinsic excitability and maintain sparse activity [104, 105]. Ultimately, the scaling of afferent synapses

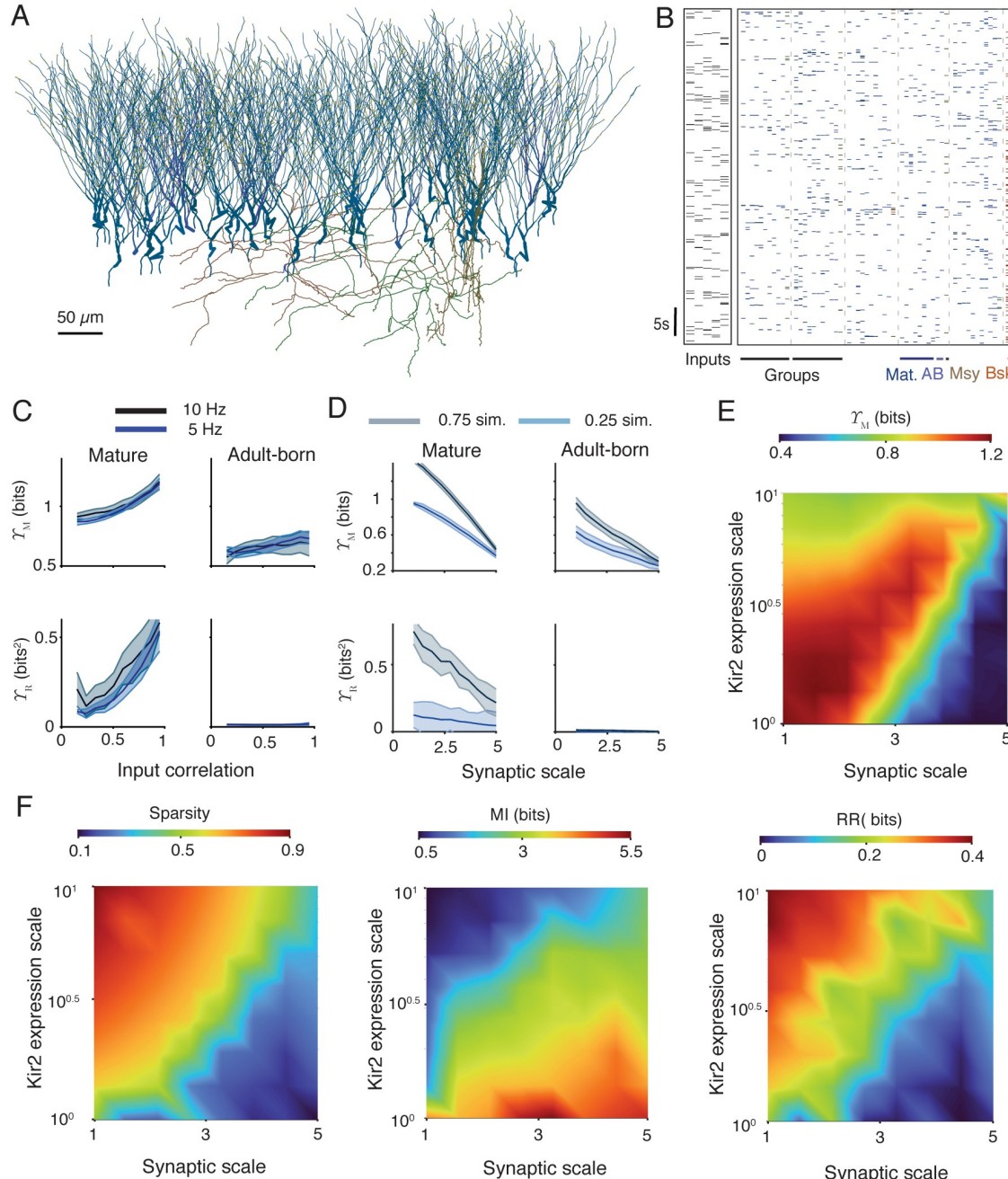

**Fig 6. Pattern separation in a network and under pathology. A** Example of dentate gyrus microcircuit. Granule cells are in shades of blue, mossy cells are in shades of brown, and the pyramidal basket cell is in red. **B** Example rasters of input patterns (left) and driven microcircuit activity (right). Vertical rows show spiking activity of a single cell and colours correspond to panel **A**. Cells are grouped into sets receiving the same inputs and sharing preferential lateral connections (see Methods). **C**. Sparsity weighted mutual information $\Upsilon_M$ (top) and relative redundancy reduction $\Upsilon_R$ (bottom) for the mature (left) and adult-born (right) granule cell populations as a function of input phase-locked correlation strength. Solid lines represent the mean over 10 repetitions and the shaded areas show one standard deviation above and below the mean. Blue shows the response to input ensembles with a spiking rate of 5Hz and grey to input ensembles at 10Hz. Spike traces are two minutes long and consist of phase-locked inputs with a phase rate of 0.6Hz. **D**. Sparsity weighted mutual information $\Upsilon_M$ (top) and relative redundancy reduction $\Upsilon_R$ (bottom) for the mature (left) and adult-born (right) granule cell populations as a function of a scaling parameter for the informative lateral perforant path synapses. Solid lines represent the mean over 10 repetitions and the shaded areas show one standard deviation above and below the mean. Blue shows a weak input similarity (phase-locked correlation strength of 0.25), and grey a strong input similarity (phase-locked correlation strength of 0.75). Spike traces are two minutes long and consist of phase-locked inputs with a phase rate of 0.6Hz and a spiking rate of 5Hz. **E** Simulations of epilepsy-related changes in synaptic input weights (horizontal axis) and Kir2 channel density (vertical axis).

Sparsity weighted mutual information $\Upsilon_M$ as a joint function of synaptic and Kir2 expression scaling parameters. The heatmap shows the average over ten repetitions. Spike traces are two minutes long and consist of phase-locked inputs with a strength of 0.75, a phase rate of 0.6Hz, and a spiking rate of 5Hz. The relative redundancy reduction $\Upsilon_R$ is plotted in S5(B) Fig. **F** Components of pattern separation as a joint function of synaptic and Kir2 expression scaling parameters. From left to right: sparsity $(m_X - n_Y)/m_X$, mutual information $I_{X,Y}$, and redundancy reduction $R_X - R_Y$ (see Eqs 1 to 3).

onto adult-born granule cells [106] and the extent of their integration into dentate gyrus microcircuits [12] will determine their contribution to pattern separation in the brain.

## Pattern separation under pathology

Next we provide a proof of principle and demonstrate the usefulness of our information theoretical measures for assessing the function of the dentate gyrus under pathological conditions, when changes to input codes and pattern completion mechanisms are potentially unknown. For this purpose, we apply our new measures to estimate the performance of the above network under the types of changes associated with epilepsy. Epilepsy causes structural changes in the dentate gyrus that increase granule cell activity and reduce behavioural discrimination performance [43]. A notable structural change is a significant increase in the sizes of axonal boutons and dendritic spine heads at perforant path synapses, and an associated increase in synaptic strength [40]. This is likely to hinder pattern separation by increasing the reliability of input responses by each principal cell [39]. In agreement with this expectation, scaling up the informative synapses to the granule cells in the above networks shows a decrease in pattern separation performance by both $\Upsilon_M$ and $\Upsilon_R$ (Fig 6D).

Another change in the epileptic dentate gyrus, is an increase in the expression of inward rectifying potassium (Kir2) channels in granule cells [38]. This mechanism increases the conductance of granule cells and so reduces their excitability, potentially compensating for increased afferent synaptic strength. Measuring $\Upsilon_M$ as a function of scaling parameters for both the Kir2 channel densities and the informative synaptic strengths, reveals a region where pattern separation performance could be maintained by upregulating Kir2 expression to counteract increased synaptic weights (Fig 6E). To see exactly how this occurs, Fig 6F plots how sparsity, pure mutual information, and pure redundancy reduction change as a joint function of the synaptic scale and Kir2 conductance scale. Sparsity and redundancy reduction decrease with synaptic strength, but increase with Kir2 conductance. Mutual information has a largely inverse relationship to the other two measures, but not so much as to cancel out the region of higher pattern separation seen in Fig 6E. These results show that the new information theoretic measures of pattern separation produce reliable and intuitive results when applied to complex compartmental models that seek to simulate real systems.

## Pattern separation in a large network

The pattern separation measures defined here are not limited to small networks and can be scaled up to larger systems. In Fig 7 we simulate networks of 50, 000 and 100, 000 granule cells receiving 5, 000 or 10, 000 two minute long inputs that are generated from a grid cell-type firing pattern (Fig 7A). The granule cells are simplified to point neurons with adaptive exponential integrate-and-fire spiking properties fitted to the compartmental models above [107, 108]. In this model it is possible to explore a number of network features at larger scale and directly compare the performance of the sparsity-weighted mutual information $\Upsilon_M$ against a classical measure, the Hamming distance $\Upsilon_\eta$, for networks of very large size.

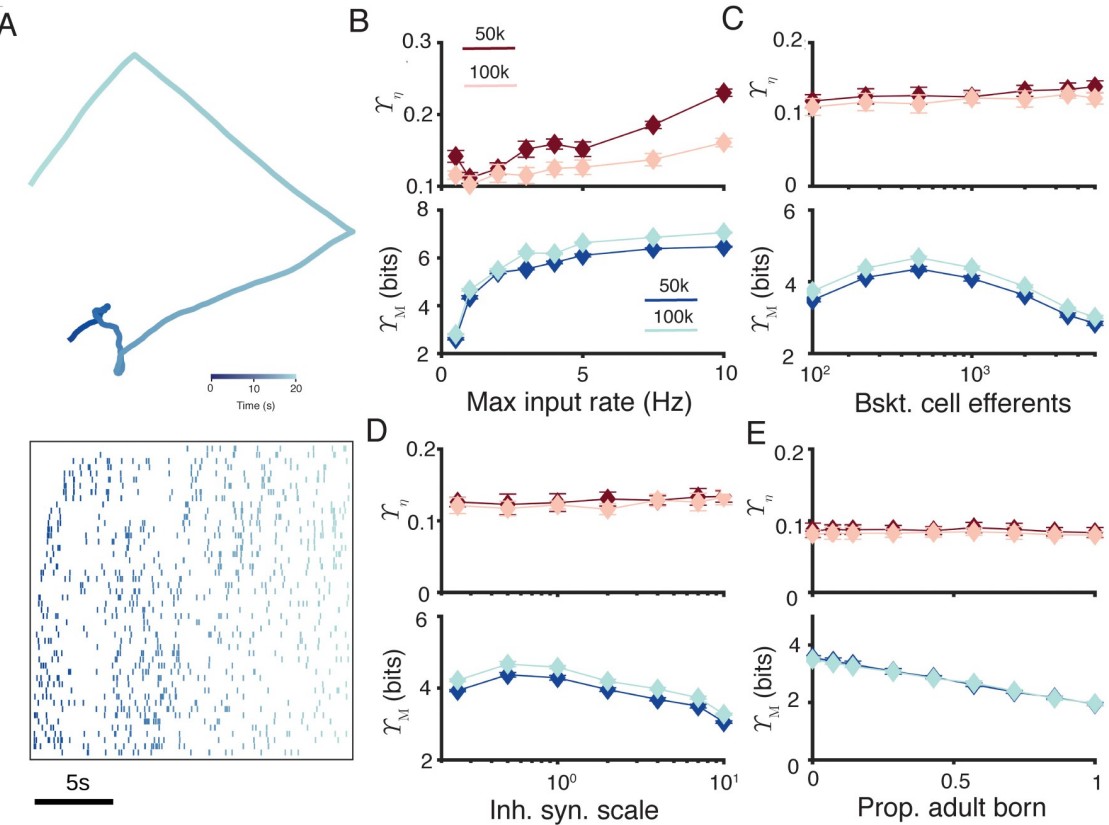

**Fig 7. New pattern separation measures can be applied to larger network models. A** Example of inputs to the large network model. Top. Example path in space that generates grid cell-like firing. Colour gradient indicates time over 20s. Bottom: Input firing rasters of 50 neurons over the shown path for 20s. Colours match the time above. **B** Pattern separation as measured by Hamming distance $\Upsilon_\eta$ (top) and sparsity-weighted MI $\Upsilon_M$ (bottom) as a function of the maximum firing rate of input cells. Dark lines correspond to 5, 000 inputs to 50, 000 principal cells, and light lines to 10, 000 inputs to 100, 000 principal cells. Error bars show one standard deviation over 24 repetitions. **C** Pattern separation as a function of the number of principal cells innervated by each of the 100 (50k network) or 200 (100k network) inhibitory basket cells. **D** Pattern separation as a function of the synaptic strength of the inhibitory basket neurons. A scale of 1 corresponds to 1nS. **E** Pattern separation as a function of the proportion of principal neurons that are adult-born. Other parameters are as described in Methods and colours and markers are as described in **B**.

We varied the maximum firing rate of the inputs, and found an increase in $\Upsilon_M$ with firing rate (Fig 7B, using a population rate code), but not consistent changes in $\Upsilon_\eta$ (all brown lines in Fig 7) at this scale. When varying the number of principal neurons innervated by each of the 100 or 200 inhibitory basket cells we found an intermediate peak in $\Upsilon_M$ (Fig 7C), matching the idea that a balanced level of recurrent inhibition is best for pattern separation. A similar result was found for up- and -downscaling the strength of the inhibitory synapses from the basket cells to each of their 500 efferent granule cells 7D). Finally, increasing the proportion of adult-born granule cells led to a linear decrease in pattern separation performance 7E). This result is consistent with our detailed compartmental simulations in Fig 6C, but again may change if the integration of the adult-born cells into the feedforward and lateral connectivity of the network is altered according to experimental data on their synapses as the adult-born cells provide another source of network heterogeneity [12, 106]. There is also likely to be some interaction between the proportion of more excitable adult-born cells and the optimal level of recurrent inhibition in Fig 7C and 7D.

Interestingly, doubling the size of the network from 50, 000 to 100, 000 principal cells consistently leads to a slight increase in $\Upsilon_M$ for all network parameters (dark blue lines against light blue lines in Fig 7). This is driven by a slight increase in the mutual information of the ensemble codes with size.

## Discussion

Classical approaches to measuring pattern separation typically assume some form of pattern completion [20, 54], and the performance of a circuit in pattern separation is intrinsically tied to the complementary completion circuit; one separation-completion network performs better than another if it has greater storage capacity and higher tolerance to noise. Taking classical measures such as orthogonalisation, decorrelation, or increased spike train distances in isolation, however, leaves them vulnerable to confounding improved pattern separation performance with loss of information. Whilst healthy biological systems may be typically assumed to maintain information flow [58], this cannot be assumed when assessing pathologies such as epilepsy [38], or when artificially manipulating the system in question [109]. We have shown explicitly that classical techniques can suffer from an unavoidable parameter dependence that makes drawing conclusions about their meaning difficult, and that they indeed cannot distinguish pattern separation from pattern destruction. The major goal of this study is to describe and promote the application of tools from information theory, in particular the sparsity-weighted mutual information and relative redundancy reduction, to address these issues and to consistently and reliably quantify the ability of the dentate gyrus to separate input patterns whilst maintaining information throughput. The components of our measures echo the two perspectives taken by Aimone et al [63] and Sahay et al [64] on the key features of dentate gyrus function, decreasing similarity and maintaining information flow. Processes that decrease input similarity typically decrease information flow; we believe that both must be considered together to truly measure pattern separation. We have demonstrated our new techniques on single cell models and shown that they allow more useful conclusions to be drawn about the contribution of synaptic plasticity to pattern separation performance than classical measures do. We have demonstrated in an abstract autoassociative network that our new measures can be predictive of pattern completion accuracy. We have further shown that our information theoretic techniques allow nuanced analysis of the different pattern separation behaviours of mature and adult-born granule cells in a network model, and that they can be used to study network pathology and mechanisms that might maintain functional homeostasis under epileptic changes.

There is space to develop the ideas of information theory for pattern separation further. The spiking codes here, either explicitly or implicitly in the case of the estimators $\tilde{\Upsilon}_M$ and $\tilde{\Upsilon}_R$, do not assume a functional form for the probability distributions over the coding sample space and infer these directly from the data. In reality, there is a body of experimental and theoretical work on the expected distributions of activity in hippocampal cells. Leutgeub et al [110], for example, showed sparse yet overlapping granule cell activity in response to changing spatial environments, and Mizuseki and Buzsáki [111], found conserved lognormal distributions of firing rates over both dentate gyrus granule cells and hippocampal pyramidal cells. Incorporating knowledge about the expected distributions of activity could allow conclusions about contributions to pattern separation to be refined based on factors such as the assumed spatial tuning of some granule cells, their different intrinsic excitabilities, or the frequency of input signals [112, 113]. When pattern separation performance is believed to change over time, the rate of information transmission at a given time (in bits/s) could be calculated instead of the total mutual information (in bits) in our new measures. For specific spiking codes, it is also

possible to identify precisely which parts of an input ensemble are redundant, and if they are removed by a pattern separation circuit.

There is also the idea that neural codes may be multiplexed, with the same sets of spikes carrying multiple distinct information streams [52, 114]. For example, the spike rate might encode one stimulus and the spike times within that rate might encode another. When a signal is believed to be multiplexed, it would be possible to extend our information theoretic measures to capture the information found in different codes separately. The 'indexing' theory of hippocampal memory [115, 116] is also worth mentioning here; if the role of the dentate gyrus is to simply pass a unique index to CA3 for a specific pattern of cortical activity, then mutual information alone would be a better measure of performance than either the weighted versions here or any classical measure. The dentate gyrus might also play a role in 'data stirring' [117], generating noise to allow neural representations to be better distributed between cell assemblies in the pattern completion circuit. In this case, the question arises as to whether information passes through the dentate gyrus at all or through direct connections from the entorhinal cortex to the hippocampus proper [1]. In the latter case it would be necessary to apply pattern separation measures, whether classical or information theoretic to the entirety of the outputs from the entorhinal cortex and the entirety of the inputs to CA3 and CA2.

The role of adult-born granule cells in pattern separation remains controversial, there are demonstrated mechanisms by which they might contribute to both sparsity [64] and information throughput [59, 63]. Implementing experimentally determined weights and densities for modelled synapses onto adult-born granule cells [105, 106, 118] and adding details about the integration of adult-born granule cells into the dentate gyrus microcircuitry [12, 119–121] would facilitate the assessment of their specific contribution to pattern separation using our new metrics.

Pattern separation is not a phenomenon associated only with the dentate gyrus. The cerebellar cortex and mushroom body both also contain circuits believed to differentiate input patterns, but with different constraints on information throughput and different underlying mechanisms [61]. Directly estimating the information content of separated patterns in these different circuits could allow the roles of different circuit components to be better understood, and a holistic theory of what precisely is necessary for pattern separation to be developed. An interesting extension of this work would be to use the fact that mutual information can be computed between sample spaces of different types to directly link pattern separation at the circuit level to behavioural discrimination. Although Santoro [34] cautioned against the usage of the phrase 'pattern separation' to refer to behaviour, there are established links between circuit changes in the dentate gyrus and behavioural outcomes [37, 38, 41–43]; computing information between circuit activities and behavioural states would be an interesting step with potential clinical relevance.

A precise assessment of pattern separation performance is particularly important for population modelling of hippocampal cells and circuits and assessment of optimality (cf. [122]). Many studies have shown that ion channel distributions can be highly degenerate, with many different parameter combinations leading to similar functional behaviour [123–125]. It remains an open question how well different potential combinations can be constrained by further considerations, such as energy consumption or efficiency [126, 127]. A promising idea in recent studies is the use of Pareto optimality to reduce and better understand the spaces of parameters and functionalities spanned by populations of neurons [122, 128, 129]. The assumption here is that evolution will tend to favour neurons and circuits whose performance at one possible task cannot be improved without reducing their performance at some other tasks; such neurons would be Pareto optimal. Neuronal systems that are fully optimal in a given situation, using a specific balance of tasks, must then be drawn from the set of Pareto

optimal cells. Effective application of these principles to population models of neurons requires a robust and consistent way to assess their performance on different tasks. For neurons believed to perform pattern separation, assessment of their functionality will be far more reliable with the information theoretic measures introduced here than with previous techniques.

We hope that the techniques highlighted here will become a standard way to quantify the ability of a neuron or circuit to perform pattern separation, particularly when that neuron is isolated from its pattern completion mechanism, or affected by pathology or experimental manipulation. Our new techniques can be applied to both experimental and theoretical studies. We believe that such techniques have a natural place in assessing the functionality of pattern separation circuits and can allow for robust and nuanced conclusions to be drawn about the principles that constrain their structure and function.

## Methods and models

### Standard metrics of pattern separation

A number of measures have been used to quantify pattern separation in neuronal data. We here use five existing measures of spike train dissimilarity for comparison with our new information theoretic measures. These are designated with $\Upsilon$ and a lower-case greek subscript to differentiate them from the upper-case latin subscripts used for the information theoretic measures. In general, we consider metrics of pattern separation as the ratio of output similarity to input similarity. This minimises input similarity as an additional source of variability and keeps the measures consistent, but is different from the approach of studies such as Madar et al [52] where the difference between input and output similarities is used. Other studies consider the input and output values separately [49], their decreased overlap [14], or the values integrated over a range of different inputs [45]. The latter approach removes the influence of input similarity on the measure, but requires more simulation time. Studies such as Yim et al [39] plot both input and output similarities, whilst fitting ratios as here in order to quantify changes in pattern separation performance. Such an approach retains the benefit of presenting both input and output similarities separately for direct visual comparison, a common feature in pattern separation studies, but also provides a single metric to directly measure changes in performance. For all measures that require discretisation, a bin size of 10ms is used unless otherwise stated.

**Orthogonalisation $\Upsilon_\theta$.** Orthogonalisation can be quantified as a reduction in the cosine distance between spike trains discretised into vectors. Increasing the cosine distance between vectorised spike trains allows them to better span a given space. Orthogonalisation of spike trains allows for pattern separation in the original and most fully-developed models [130]. If $X$ and $Y$ are vectorised spike trains, the cosine distance $\theta(X, Y)$ between them is given by

$$\theta(X, Y) = \frac{X \cdot Y}{\|X\| \|Y\|} \tag{4}$$

where $X \cdot Y$ is the dot product of $X$ and $Y$ and $\|X\|$ is the euclidean norm of $X$. A cosine distance of 0 indicates perfectly orthogonal spike train vectors and a cosine distance of 1 indicates perfectly collinear spike train vectors. $\Upsilon_\theta$ is defined as the reduction in mean pairwise cosine distance between input and output spike train ensembles. If there are $N$ input spike train vectors $\{X_1, X_2, \ldots, X_N\}$ and $M$ output spike train vectors $\{Y_1, Y_2, \ldots, Y_M\}$, then

$$\Upsilon_\theta = \frac{M(M-1) \sum_{i \neq j}^{N} \theta(X_i, X_j)}{N(N-1) \sum_{i \neq j}^{M} \theta(Y_i, Y_j)} \tag{5}$$

**Scaling $\Upsilon_\sigma$.** Cosine distance (Eq 4) is defined for normalised vectors and is independent of differences in vector magnitude. We follow Madar et al [52] in defining a complementary scaling factor $\sigma(X, Y)$ between vectorised spike trains $X$ and $Y$ such that

$$\sigma(X, Y) = \begin{cases} \frac{\|X\|}{\|Y\|} & \text{if } \|X\| \leq \|Y\| \\ \frac{\|Y\|}{\|X\|} & \text{if } \|X\| > \|Y\| \end{cases} \tag{6}$$

A scaling factor of 1 indicates spike trains with the same magnitude. $\Upsilon_\sigma$ is defined as the reduction in mean pairwise scaling factors between input and output spike train ensembles. If there are $N$ input spike train vectors $\{X_1, X_2, \ldots, X_N\}$ and $M$ output spike train vectors $\{Y_1, Y_2, \ldots, Y_M\}$, then

$$\Upsilon_\sigma = \frac{M(M-1) \sum_{i \neq j}^{N} \sigma(X_i, X_j)}{N(N-1) \sum_{i \neq j}^{M} \sigma(Y_i, Y_j)} \tag{7}$$

**Decorrelation $\Upsilon_\rho$.** Pattern separation has also been defined as the reduction in pairwise correlations between input and output ensembles of spike train vectors [45]. The Pearson correlation coefficient $\rho_{X,Y}$ between vectorised spike trains $X$ and $Y$ is defined as [131]

$$\rho_{X,Y} = \frac{\mathbb{E}[(X - \mu_X)(Y - \mu_Y)]}{\sigma_X \sigma_Y} \tag{8}$$

where $\mu_X$ and $\sigma_X$ are the mean and standard deviation of $X$ respectively. $\rho_{X,Y}$ varies between $-1$ for perfectly anticorrelated and 1 for perfectly correlated spike train vectors. If there are $N$ input spike train vectors $\{X_1, X_2, \ldots, X_N\}$ and $M$ output spike train vectors $\{Y_1, Y_2, \ldots, Y_M\}$, then

$$\Upsilon_\rho = \frac{M(M-1) \sum_{i \neq j}^{N} \rho_{X_i, X_j}}{N(N-1) \sum_{i \neq j}^{M} \rho_{Y_i, Y_j}} \tag{9}$$

**Hamming distance $\Upsilon_\eta$.** A standard way to determine the distance between vectors is the Hamming distance [132]. This is simply the number of elements at which the vectors differ; the Hamming distance $\eta$ between binarised spike trains $X$ and $Y$ is defined as

$$\eta(X, Y) = \sum_{i=1}^{n} |X(i) - Y(i)| \tag{10}$$

where $X(i)$ is the $i$-th element of the spike vector $X$ of length $n$ and $|x|$ is the absolute value of $x$. If there are $N$ input spike train vectors $\{X_1, X_2, \ldots, X_N\}$ and $M$ output spike train vectors $\{Y_1, Y_2, \ldots, Y_M\}$, then

$$\Upsilon_\eta = \frac{M(M-1) \sum_{i \neq j}^{N} \eta(X_i, X_j)}{N(N-1) \sum_{i \neq j}^{M} \eta(Y_i, Y_j)} \tag{11}$$

Some studies modify the Hamming distance to account for sparsity in activity by removing permanently inactive spike trains from the ensembles [47, 49]. This adjustment does not have an impact here as there are no completely unresponsive neurons.

**Wasserstein distance $\Upsilon_\delta$.** The Wasserstein distance quantifies the minimum change in probability density needed to move between two probability distributions [133]. It has recently

been applied to neuronal spike trains by Sihn and Kim [53]. Briefly, if spike train $\mathbf{X}$ consists of spikes at times $\{x_1, x_2, \ldots, x_n\}$ and spike train $\mathbf{Y}$ consists of spikes at times $\{y_1, y_2, \ldots, y_m\}$, the Wasserstein distance $\delta(\mathbf{X}, \mathbf{Y})$ between them is given by

$$\delta(\mathbf{X}, \mathbf{Y}) = \int_{\min(x_1, y_1)}^{\max(x_n, y_m)} \left| \left( \chi_{[x_n, \infty)}(t) + \sum_{i=1}^{n-1} \frac{i}{n} \chi_{[x_i, x_{i+1})}(t) \right) - \left( \chi_{[y_m, \infty)}(t) + \sum_{j=1}^{m-1} \frac{j}{m} \chi_{[y_j, y_{j+1})}(t) \right) \right| dt \quad (12)$$

where $\chi_{[a,b)}(t)$ is the indicator function of the interval $[a, b)$. $W(\mathbf{X}, \mathbf{Y})$ is unbounded and $\Upsilon_\delta$ is defined as the reduction in mean pairwise Wasserstein distance between input and output spike train ensembles. If there are $N$ input spike trains $\{\mathbf{X}_1, \mathbf{X}_2, \ldots, \mathbf{X}_N\}$ and $M$ output spike trains $\{\mathbf{Y}_1, \mathbf{Y}_2, \ldots, \mathbf{Y}_M\}$, then

$$\Upsilon_\delta = \frac{M(M-1) \sum_{i \neq j}^N \delta(\mathbf{X}_i, \mathbf{X}_j)}{N(N-1) \sum_{i \neq j}^M \delta(\mathbf{Y}_i, \mathbf{Y}_j)} \quad (13)$$

Unlike the above measures which map spike times to a vector, $\Upsilon_\delta$ does not require a choice of bin size. We prefer it to other popular measures of spike train distance such as those of van Rossum [134], which requires a choice of smoothing parameter, or Kreuz et al [50], which is more sensitive to the differences in rates between spike trains that are already measured by $\sigma$ (Eq 6).

## Mutual information

The mutual information $I_{X,Y}$ between two random variables $X$ and $Y$ quantifies the reduction in uncertainty about the value of one variable that knowing the value of the other would provide [71]. It takes a minimum of 0 for independent random variables and a maximum of the entropy of either random variable if the relationship is entirely deterministic. $I(X;Y)$ is defined as

$$I_{X,Y} = \sum_x \sum_y p_{(X,Y)}(x, y) \log_2 \left( \frac{p_{(X,Y)}(x, y)}{p_{(X)}(x) p_{(Y)}(y)} \right) \quad (14)$$

where $p_{(X,Y)}$ is the joint distribution of $X$ and $Y$ and $p_X$ and $p_Y$ are their respective marginals.

For neural data the distributions $p_{(X,Y)}$, $p_X$, and $p_Y$ depend on the choice of neuronal coding strategy (see 'Neuronal codes' below) and discretisation. The coding strategy and discretisation should in principle be chosen to maximise $I$ in each case.

## Transfer entropy

The transfer entropy $T_{X \to Y}$ [83] from one random variable $X$ to another $Y$ is the mutual information between $Y$ at the current time $T$ and the history of $X$ conditioned on the history of $Y$:

$$\begin{aligned} T_{X \to Y} &= H(Y_T | Y_{t<T}) - H(Y_T | Y_{t<T}, X_{t<T}) \\ &= \sum_x \sum_y \sum_{y_{t<T}} p_{(X,Y,Y_{t<T})}(x, y, y_{t<T}) \log_2 \left( \frac{p_{(Y_{t<T})}(y_{t<T}) \, p_{(X,Y,Y_{t<T})}(x, y, y_{t<T})}{p_{(X,Y_{t<T})}(x, y_{t<T}) \, p_{(Y,Y_{t<T})}(y, y_{t<T})} \right) \end{aligned} \quad (15)$$

where $H(X|Y)$ is the conditional entropy of $X$ given $Y$. As for the mutual information, the distributions depend on the choice of neuronal coding strategy (see 'Neuronal codes' below) and discretisation, but these can be chosen to maximise the measured entropy. Unlike mutual information, transfer entropy is directional and asymmetric between inputs and outputs. In

many circumstances, the transfer entropy requires less data to produce an accurate estimate than the mutual information [86, 89].

## Redundancy

Redundancy means that multiple parts of a signal may encode the same information [78, 84, 92, 135]. To use this idea to asses pattern separation, we adapt the partial information decomposition of Williams and Beer [84] to spike trains in an ensemble. In particular, if $X$ is a spike train ensemble made up of trains $\{x_1, x_2, \ldots, x_n\}$ we consider the redundancy of $X$ to be the minimum mutual information between any individual train $x_i$ and the remainder of the ensemble $X \backslash x_i$

$$R_X = \min_{x_i} \sum_{x \in X \backslash x_i} \left( \sum_{y \in x_i} p_{(X \backslash x_i, x_i)}(x, y) \log_2 \left( \frac{p_{(X \backslash x_i, x_i)}(x, y)}{p_{(X \backslash x_i)}(x) p_{(x_i)}(y)} \right) \right) \tag{16}$$

Again, the probability distributions depend on the choice of neuronal code. Redundancy can be applied to both the input and output ensembles, and these are combined to give the pattern separation measure redundancy ratio $\Upsilon_R$ (Eq 3).

## Neuronal codes

The joint and marginal distributions over the spike trains depend on which features are considered. We use the common terminology of 'words' to refer to the significant patterns that make up the sample spaces over which information is calculated; 'words' can be formed from different combinations of 'letters'. All distributions are computed empirically from the data, without assuming a specific functional form. It is not necessary for input and output spike train ensembles to use the same coding strategy, so informational measures are maximised over all possible input-output pairs.

**Instantaneous spatial code.** The instantaneous spatial code assumes that information is encoded by the current state of all spike trains in the ensemble. Spike trains are binarised into bins and 'words' are formed from the state of the bins in each train at a given time. If there are $n$ trains in the ensemble, there are therefore $2^n$ possible 'words'.

**Local temporal code.** The local temporal code assumes that information is encoded by spike times in each train individually. Spike trains are binarised into bins and 'words' are formed from the state of the consecutive bins in a given train. If the word length is $w$, there are $2^w$ possible words.

**Local rate code.** The local rate code assumes that information is encoded by the instantaneous firing rates in each train individually. Spike trains are discretised additively into bins in a given train. If the maximum number of spikes found in any bin is $r$, the number of words is $r + 1$.

**Ensemble rate code.** The ensemble rate code assumes that information is encoded by the total firing rates across all trains in the ensemble. Spike trains are discretised additively into bins and these are added across trains. If the maximum number of spikes found in any bin is $r$, the number of possible words is $r + 1$.

**Specific rate code.** The specific rate code assumes that information is encoded by the pattern of instantaneous firing rates across all trains in the ensemble. It is distinct from the ensemble rate code as the identity of the neuron firing at each rate is significant. Spike trains are discretised additively into bins and these are formed into 'words' by combining bins at a given time. If there are $n$ trains in the ensemble and the maximum number of spikes found in any bin is $r$, the number of possible words is $(r + 1)^n$.

**Spatiotemporal code.** The spatiotemporal code assumes that information is encoded by the pattern of spike times across all trains in the ensemble. Spike trains are binarised into bins and 'letters' are formed from the state of the consecutive bins in each train. These 'letters' are formed into 'words' by combining sets of bins at a given time. If there are $n$ trains in the ensemble and the 'letter' length in each train is $w$, the number of possible words is $2^{wn}$.

## Estimating mutual information and redundancy

Mutual information can also be estimated using a variety of standard techniques [136]. We demonstrate the application of such techniques using the modified Kozachenko-Leonenko estimator [85] described in Houghton [76]. This estimator exploits the proximity structure of spike trains to reduce the amount of data necessary to reliably estimate the mutual information between them. Briefly, the input and output spike train ensembles are each divided into $N$ periods of equal size. The pairwise spike train distances between each period of each spike train are computed using the Wasserstein metric $\delta$ (Eq 12) and a euclidean norm is taken over the trains in the ensemble. This produces a set of pairwise distances between periods of the input and output ensembles. A biased estimator $\tilde{I}(h)_{kl}$ of the mutual information between input and output ensembles in terms of an integer smoothing parameter $1 \leq h < N$ is given by

$$\tilde{I}_{kl}(h) = \frac{1}{N} \sum_{i=1}^{N} \log_2 \left( \frac{N}{h^2} \#\left( C_h(u_i, v_i) \right) \right) \tag{17}$$

where $\#(C_h(u_i, v_i))$ counts the number of pairs $(u_j, v_j)$ such that both $\delta(u_i, u_j) < \Delta_h(u_i)$ and $\delta(v_i, v_j) < \Delta_h(v_i)$, where $\Delta_h(u_i)$ is the distance of $u_i$ to its $h$-th nearest neighbour amongst all segments $u$. The bias $\tilde{I}_0$ can estimated as

$$\tilde{I}_0(h) = \frac{1}{N} \sum_{i=1}^{N} \frac{\binom{h-1}{i-1}\binom{N-i}{N-h}}{\binom{h-1}{n-1}} \log_2 \left( \frac{iN}{h^2} \right) \tag{18}$$

We follow Houghton [76] in choosing the smoothing parameter $h$ to maximise the unbiased estimator $\tilde{I}(h) = \tilde{I}_{kl}(h) - \tilde{I}_0(h)$, specifically.

$$\tilde{I} = \underset{h}{\operatorname{argmax}}(\tilde{I}_{kl}(h) - \tilde{I}_0(h)) \tag{19}$$

The original implementation of this technique assumed a fixed period of division, leading to a fixed value of $N$ if the total length of the spike train ensemble is constant. We relax this assumption by adapting the technique in Strong et al [75] to estimate the limiting mutual information for infinitely long spike trains. Briefly, $\tilde{I}$ is calculated for different values of $N$ and values are considered reliable if the optimal smoothing parameter $h$ is much lower than its maximum possible value of $N - 1$ (S5(A) Fig, top panels). Estimates in this region are a linear function of the number of periods $N$. The linear fit is then extrapolated back to a value of $N = 0$ to produce a consistent estimate of the total mutual information between the spike train ensembles (Fig 4, right panels, and S5(A) Fig, bottom panels).

The procedure to estimate the redundancy between trains in an ensemble is similar. As in Eq 16, we compute the minimum estimated mutual information between each train in the ensemble and the remainder taken together. This is extrapolated to the limiting mutual information as above before the minima are taken.

## Input spike trains

Ensembles of input spike trains are generated using a number of different techniques to produce different correlation structures across and within different trains. The different ensembles possess positive and negative correlations both across and within distinct trains [137, 138]

- *Phase-locked*. Phase-locked ensembles are generated using an inhomogeneous Poisson process with the instantaneous firing rate varying sinusoidally in time and constant across spike trains. This generates sets of spike trains that are periodic in time and positively correlated between different trains. Key parameters are the strength of the phase-locking, the relative amplitude of the sinusoid about the mean single neuron firing rate, and the period of the sinusoid.

- *Auto-correlated*. Spike trains are generated independently using a renewal process with gamma-distributed interspike intervals. The shape parameter $\alpha$ of the gamma distribution governs the autocorrelation of the trains. Shape parameters less than one produce bursty trains, a shape parameter of one is equivalent to an uncorrelated Poisson process, shape parameters greater than one produce periodic trains. The initial spike time in each train is generated by a Poisson process to eliminate cross-correlations.

- *Cross-correlated*. Spike trains are produced by uniformly filtering a master Poisson process with rate equal to the desired rate multiplied by the product of the number of trains in the ensemble and the correlation strength.

  Examples are plotted in S1(A) Fig.

## Granule cell models

We conduct our simulations using the morphologically robust compartmental models of mouse dentate gyrus granule cells introduced by Beining et al [94]. In brief, these models distribute the ion channels reported in the literature over morphologies based on those reconstructed by Schmidt-Hieber et al [139]. The ion channels are section specific to the soma, axon hillock, axon, granule cell layer dendrites, inner molecular layer dendrites, middle molecular layer dendrites, and outer molecular layer dendrites. Full details can be found in the original paper.

Synaptic inputs from the perforant path undergo short-term plasticity. This is modelled using the quantal version of the Tsodyks-Markram model [140], where individual vesicles undergo probablistic release and replenishment. The standard parameters are adapted from those fitted in Madar et al [44] to the lateral perforant path. The initial release probability is 0.325, there are 40 vesicle release sites per synapse, the timescale of recovery from depression is 500ms, and the timescale of facilitation is 9ms. The excitatory postsynaptic conductance change induced by a single vesicle is 0.3nS. The synapse is formed by four anatomical contacts in the outer molecular layer (see Fig 3A). In addition, each cell receives background noise from weaker synapses, these are distributed randomly with densities of 0.05, 0.1, 0.2, and 0.1 contacts per $\mu$m in the granule cell layer, inner molecular layer, middle molecular layer, and outer molecular layer respectively. Background synapses undergo independent Poisson activation with a rate of 0.5Hz and a strength of 0.01nS. All excitatory synapses are modelled as a difference of exponentials, with a rise timescale of 0.2ms, a decay timescale of 2.5ms, and a reversal potential of 0mV.

The original model assumes that ion channel conductances are constant within each section of the tree (granule cell layer, inner molecular layer, etc). For the bottom panels of Fig 3B we vary the conductance randomly within each section. Conductances in each $1\mu$m segment are

randomly drawn from a non-negative gamma distribution with mean equal to the fitted value and a coefficient of variation (ratio of standard deviation to mean) that is used to vary the spatial heterogeniety as seen in the figure. Specifically, if the mean conductance of an ion channel in a layer is $\mu$, and the desired coefficient of variation (heterogeneity) is $c > 0$, then the conductance of that ion channel in each $1\mu$m section is drawn independently from

$$f(x) = \frac{(\mu c^2)^{-\frac{1}{c^2}}}{\Gamma(c^{-2})} x^{\left(\frac{1}{c^2}-1\right)} e^{\left(-\frac{x}{\mu c^2}\right)} \tag{20}$$

where $\Gamma(z)$ is the gamma function.

A typical set of input spike patterns and corresponding output voltage traces are plotted in S4(A) Fig.

## Pattern completion

In Fig 5 we give a proof-of-principle demonstration as to how our novel measures of pattern separation would affect pattern completion in an abstract auto-associative Hopfield network [99]. The network consists of 784 recurrently-connected artificial binary neurons that each have a state $s_i \in \{-1, 1\}$. The network is initiated with a given pattern and evolves asynchronously, so that individual neuron states are updated randomly until the network has converged to a final state. Updates occur using the rule

$$s_i^+ = \begin{cases} 1, & \text{if } \sum_{j=1}^{784} w_{ij} s_j^- > 0 \\ -1, & \text{otherwise} \end{cases} \tag{21}$$

where $s_j^-$ is the state of neuron $j$ immediately before the update, $s_i^+$ is the state of neuron $i$ immediately after the update, and $w_{ij}$ is the connection weight from neuron $j$ to neuron $i$. Weights are calculated for an exemplar pattern of each of the 49 classes (S5(B) Fig) using the pseudo-inverse training rule [102]:

$$w_{ij} = \frac{1}{784} \sum_{v=1}^{49} \sum_{\mu=1}^{49} \xi_i^v \left(\mathbf{Q}_{v\mu}^{-1}\right) \xi_j^\mu \tag{22}$$

where $v$ and $\mu$ index the 49 patterns, $\xi_i^v$ is the $i$-th element of training pattern $v$, and $\mathbf{Q}_{v\mu}$ is the overlap

$$\mathbf{Q}_{v\mu} = \frac{1}{784} \sum_{k=1}^{784} \xi_k^v \xi_k^\mu \tag{23}$$

The noisy transformation is applied to patterns by selecting $m$ pixels uniformly randomly. These pixels then take a uniform random value in $\{-1, 1\}$. For each transformation, the locations of the randomised pixels are fixed, but their values are generated independently for each pattern. The correlation $\rho$ between two different pattern classes is calculated as the average between all patterns in the classes. The mutual information for each class is calculated between the example pattern used to train the network and the set of all other patterns in the class. The redundancies are calculated between the different classes.

## Network models of detailed neurons

In Fig 5 we consider a small network of excitatory and inhibitory cells from the the mouse dentate gyrus. The network is randomly generated for each repetition, with potentially different morphologies and connectivites. The principal cells are 40 mature and 10 adult-born granule

cells adapted from the single cell model described above. Heterogeneity is included by using the range of available morphologies introduced in Beining et al [94] and varying the channel densities with a coefficient of variation of 0.1 for the mature and 0.2 for the adult-born cells. 5 mossy cells and 1 inhibitory pyramidal basket cell are modelled as integrate-and-fire units with a membrane time constant of 15ms.

Connectivity for the microcircuit broadly follows the principles described in Hainmueller and Bartos [5] and employs motifs believed to support pattern separation as discussed by Cayco-Gajic and Silver [61].

- *Perforant path to granule cells*. These synapses are as described in the single cell case above, with informative synapses innervating the outer molecular layer and displaying quantal short-term plasticity. Background synapses randomly innervate the rest of the dendritic tree, with a fixed independent likelihood based on dendritic length. There are no differences in synaptic structure or parameters between adult-born and mature granule cells. Groups are formed by sets of 8 mature and 2 adult-born cells receiving the same informative presynaptic spikes, but with independent stochastic synaptic dynamics. This provides unreliable expansion to a larger coding space.

- *Granule cells to mossy cells*. Granule cells form excitatory connections to mossy cells. Each group of granule cells is preferentially recurrently connected to a single mossy cell, with probabilities of cells inside the group contacting the mossy cell being 0.9 each, and 0.025 for innervating each of the other mossy cells. The mossy cells act as a second threshold.

- *Mossy cells to granule cells*. Mossy cells form lateral excitatory connections to granule cells. Each mossy cell is preferentially connected to a group of granule cells, with probability 0.9 of contacting each cell inside the group and 0.01 of contacting each granule cell outside the group. Synapses are located at the soma of the granule cell and have a conductance of 0.5nS.

- *Granule cells to pyramidal basket cells*. Mature granule cells have the potential to form excitatory connections to the pyramidal basket cell and do so with probability 0.5. Adult-born granule cells do not form these connections. This motif provides competitive inhibition between groups of principal cells.

- *Pyramidal basket cells to granule cells*. The pyramidal basket cell forms lateral inhibitory connections to all granule cells, including the adult-born population. Synapses are located at the granule cell somata and have a conductance of 1nS, a double exponential timecourse with rise and decay timescales of 0.5ms and 5ms respectively, and a reversal potential of −80mV.

Further connections, such as direct granule cell recurrency, which is observed in primates but not healthy rodents [5], are not included.

**Pathological changes to the network model.** To model some of the effects of epilepsy, two changes are made to the above network model for Fig 5D and 5E. Firstly, informative synapses are scaled up by factors from 1 to 5 by increasing the conductance induced by each individual vesicle, to cover the range of structural changes measured in Janz et al [40]. Secondly, the conductances of all inward rectifier K$^+$ (Kir2) channels are scaled up by factors from 1 to 10 to cover the range of conductance changes measured in Young et al [38]. All other network features remain as above.

## Network models of point neurons

In Fig 7 we expand our detailed model to consider much larger networks and so reduce the fidelity of our simulations. The compartmental models of mature and adult-born granule cells

are replaced with single-compartment adaptive exponential integrate-and-fire neurons [107, 108]. The voltage $V$ of each neuron evolves as

$$\tau \frac{dV}{dt} = E_v - V + \Delta_T \exp\left(\frac{V - \theta_T}{\Delta_T}\right) + R(I(t) - w(t)) \tag{24}$$

where $\tau$ is the neuron time constant, $E_v$ is the reversal potential, $\Delta_T$ is the sharpness of action potential initiation, $\theta_T$ is the 'threshold', $R$ is the somatic input resistance, $I(t)$ is the input current, and $w(t)$ is an adaptation variable governed by

$$Fig\tau_w \frac{dw}{dt} = a(V(t) - E_v) - w + b \sum_{spikes} \delta(t - t_{spike}) \tag{25}$$

where $\tau_w$ is the adaptation time constant, $a$ is the adaptation coupling strength, $b$ is the increase in $w$ caused by a single spike, and $\delta$ is a Dirac delta function. The voltage in Eq 24 goes to infinity in finite time once it exceeds $\theta_T$, so a spike is recorded once the voltage exceeds 0mV and the voltage is reset to $E_v$.

To match the responses of the point neurons to the full compartmental models, the parameters of Eqs 24 and 25 are fitted to each cell in the population of models using the dynamic IV-curve method described by Badel et al [141] for threshold properties (S6(A) Fig) and the pattern search global optimisation algorithm for the remaining adaptation properties (S6(B) Fig). The latencies and transfer resistances for dendritic inputs from the lateral and medial perforant paths are also estimated (S6(C) and S6(D) Fig). This produces a joint distribution of parameters across the populations of mature and adult-born neurons (S6(E) Fig), from which sample neurons can be drawn to reflect the statistics and heterogeneity of the full compartmental models [142].

Unless otherwise stated, connectivity is as above. Groups of 10 granule cells, initially 8 mature and 2 adult-born receive common inputs. Inputs are generated with grid-cell like firing patterns (see below). Basket cells are now much rarer, with 100 basket cells in the 50, 000 granule cell model and 200 in the 100, 000 granule cell model. This better reflects their proportions in the real dentate gyrus [5]. Synapses to basket cells are therefore weakened by a factor of ten, and each basket cell typically makes 500 contacts onto granule cells.

**Grid cell-like inputs.** Inputs to granule cells in the larger model are based on grid cell firing. Each input cell fires as an inhomogeneous Poisson process with an instantaneous rate depending on the location of a particle making a directed random walk (Fig 7A). Each input cell $i$ responds to the position of the particle $\mathbf{x}(t)$ relative to a randomly shifted hexagonal grid (S6(F) Fig). Instantaneous firing rates $r_i(t)$ are given by

$$r_i(t) = r_{\max} \sum_j e^{-\frac{\|\mathbf{x}(t) - \mathbf{c}_{i,j}\|^2}{\rho^2}} \tag{26}$$

where $r_{\max}$ is the maximum firing rate of a centre (typically $r_{\max} = 1$), $j$ indexes over the centres of grid firing at locations $\mathbf{c}_{i,j}$, and $\rho$ is a radius parameter (typically $\rho$ is half the distance between centres). In Fig 7, the centres are 10% of the domain size apart, leading to ~100 grid centres.

## Supporting information

**S1 Fig.** **A** Example rasters resulting from different methods of generating ensembles. All ensembles have a mean spike rate of 2.5Hz per neuron. Clockwise from top left: *Phase-locked* ensemble with a phase-locking strength of 0.9 and a phase rate of 0.6Hz, *periodic* ensemble

with $\alpha = 10$, cross-correlated ensemble with a similarity strength of 0.8, and *bursty* ensemble with $\alpha = 0.5$. **B** Filtering of spike train ensembles. From top to bottom: *random* filtering with $p = 0.5, 0.75, 0.85, 0.95$, *n-th pass* filtering with $n = 2, 4, 7, 20$, *refractory* filtering with $t = 0.18$, 0.54, 1.01, 3.71 s, and *competitive* filtering with $t = 3.4, 13.7, 17.9, 31.4$ ms. Filtering parameters are chosen to give roughly equal numbers of spikes across different filters. **C** Unnormalised orthogonalisation $\Upsilon_\theta$ values for comparison with Fig 1B. **D** Unnormalised scaling $\Upsilon_\sigma$ values for comparison with Fig 1B. **E** Unnormalised decorrelation $\Upsilon_\rho$ values for comparison with Fig 1B. **F** Unnormalised Hamming distances $\Upsilon_\eta$ values for comparison with Fig 1B. **G** Unnormalised orthogonalisation $\Upsilon_\theta$ values for comparison with Fig 1C. **H** Unnormalised scaling $\Upsilon_\sigma$ values for comparison with Fig 1C. **I** Unnormalised decorrelation $\Upsilon_\rho$ values for comparison with Fig 1C. **J** Unnormalised Hamming distances $\Upsilon_\eta$ values for comparison with Fig 1C. **K** Unnormalised Wasserstein distances $\Upsilon_\delta$ values for comparison with Fig 1C.
(TIF)

**S2 Fig.** **A** Unnormalised mutual information values using the spatial code for comparison with Fig 1D. **B** Unnormalised mutual information values using the temporal code for comparison with Fig 1D. **C** Unnormalised mutual information values using the local rate code for comparison with Fig 1D. **D** Unnormalised mutual information values using the ensemble rate code for comparison with Fig 1D. Solid, dashed, and dotted lines in all panels refer respectively to strong, medium, and weak input similarities (see Methods). **E** Scatter plot of sparsity against different classical measures of pattern separation for comparison with Fig 1E. **F** Scatter plot of sparsity against mutual information computed using different codes for comparison with Fig 1E. Different markers correspond to different methods of ensemble generation: circles are for phase-locked ensembles, squares are for periodic ensembles, diamonds are for bursty ensembles, and triangles are for cross-correlated ensembles. **G** Density map corresponding to scatter points in **E**. **H** Density map corresponding to scatter points in **F**. Density values are logarithmic. There is a consistent negative relationship between the proportion of spikes that pass through the filter and pattern separation metrics. In the four cases, the linear correlations between the proportion of spikes passing through the system and the different measures are $\Upsilon_\theta : -0.33$, $\Upsilon_\rho : -0.071$, $\Upsilon_\eta : -0.68$, and $\Upsilon_\delta : -0.28$. Conversely, the typical relationship between mutual information and the proportion of spikes is positive (Fig 1E), with linear correlations of 0.96 for the spatial code, 0.97 for the temporal code, 0.07 for the local rate code, and 0.73 for the ensemble rate code.
(TIF)

**S3 Fig.** **A** Unnormalised sparsity weighted mutual information $\Upsilon_M$ for comparison with Fig 2A. **B** Unnormalised sparsity weighted transfer entropy $\Upsilon_T$ for comparison with Fig 2B. **C** Unnormalised relative redundancy reduction $\Upsilon_R$ for comparison with Fig 2C. From top to bottom, the codes are spatial, temporal, rate local, and rate ensemble in all cases.
(TIF)

**S4 Fig.** **A** Example input spiking rasters (top) and output voltage traces (bottom) for the single granule cell model. **B** Classical pattern separation measures applied to simulated voltage traces from a granule cell model with varied timescales of synaptic depression. From left to right: orthogonalisation $\Upsilon_\theta$, scaling $\Upsilon_\sigma$, decorrelation $\Upsilon_\rho$, Hamming distance $\Upsilon_\eta$, and Wasserstein distance $\Upsilon_\delta$. Solid lines correspond to a strong input similarity and a binning window (where applicable) of 100 ms, dashed lines correspond to a strong input similarity and a binning window of 10 ms, dash-dotted lines correspond to a weak input similarity and a binning window of 100 ms, and dashed lines correspond to a strong input similarity and a binning window of 10ms. **C** As above with varied timescales of synaptic facilitation. **D** As above with varied ion

channel spatial heterogeneities. Input spike traces are two minutes long and consist of phase-locked inputs with a phase rate of 0.6Hz and a spiking rate of 5Hz.
(TIF)

**S5 Fig.** **A** Examples of extrapolating mutual information estimates. The left and right columns show two different ensembles. The top row shows the optimal smoothing parameter $h$ (Eq 19) as a function of the number of sections the spike trains are divided into $N$. The bottom panels show the relationship between $N$ and the mutual information estimate (grey circles), and the extrapolation back to infinitely long sections (black lines) from the region where $h$ is significantly less than $N$ and the relationship between $N$ and the mutual information estimate is linear. **B** Exemplar patterns for each of the 49 classes in the Kuzushiji-49 dataset [101]. **C** Relative redundancy reduction $\Upsilon_R$ for the mature granule cells as a function of scaling parameters for informative synapses and Kir2 channel expression. For comparison with Fig 5E.
(TIF)

**S6 Fig.** **A** Example dynamic IV curves for mature (dark blue) and adult-born (light blue) granule cells. Circles show the simulated results from full compartmental models and solid lines the best fit under Eq 24. **B** Example voltage traces for mature granule cells showing spike-frequency adaptation. **C** Example somatic voltages in a mature granule cell in response to synaptic inputs at increasing distances (lighter blue lines). **D** Latency of synaptic inputs at the soma (left) and transfer resistance to the soma (right) as a function of relative path length for mature (dark blue) and adult-born (light blue) granule cells. Shaded areas show one standard deviation around the mean. **E** Adaptive EIF parameter distributions for mature (lower left) and adult-born (upper right) granule cell compartmental models. The leading diagonal panels show single marginal distributions, and the off-diagonal panels show the pairwise marginals. **F** Examples of grid cell-like firing rates as a function of location (Eq 26).
(PDF)

**S1 Code. zip file containing code for the pattern separation toolbox and paper figures.**
(ZIP)

## Acknowledgments

We would like to thank Lucas Mongiat (Centro Atómico Bariloche) for critically reading and discussing this manuscript.

## Author Contributions

**Conceptualization:** Alexander D. Bird, Peter Jedlicka.

**Data curation:** Alexander D. Bird, Peter Jedlicka.

**Formal analysis:** Alexander D. Bird.

**Funding acquisition:** Peter Jedlicka.

**Investigation:** Alexander D. Bird, Peter Jedlicka.

**Methodology:** Alexander D. Bird.

**Project administration:** Alexander D. Bird, Hermann Cuntz, Peter Jedlicka.

**Resources:** Alexander D. Bird, Peter Jedlicka.

**Software:** Alexander D. Bird, Hermann Cuntz, Peter Jedlicka.

**Supervision:** Alexander D. Bird, Hermann Cuntz, Peter Jedlicka.

**Validation:** Alexander D. Bird, Hermann Cuntz, Peter Jedlicka.

**Visualization:** Alexander D. Bird, Hermann Cuntz, Peter Jedlicka.

**Writing – original draft:** Alexander D. Bird, Hermann Cuntz, Peter Jedlicka.

**Writing – review & editing:** Alexander D. Bird, Hermann Cuntz, Peter Jedlicka.

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
