## [Decision Letter · Decision Letter 0]

4 Jan 2023

Dear Dr Bird,

Thank you very much for submitting your manuscript "Robust and consistent measures of pattern separation based on information theory and demonstrated in the dentate gyrus" for consideration at PLOS Computational Biology.

As with all papers reviewed by the journal, your manuscript was reviewed by members of the editorial board and by several independent reviewers. In light of the reviews (below this email), we would like to invite the resubmission of a significantly-revised version that takes into account the reviewers' comments.

In particular, as well as responding to the reviewer's comment regarding the simulations presented here, the authors should make a concerted effort to improve the clarity of the manuscript and figures, ensuring that these new results are placed in the appropriate historical context. 

We cannot make any decision about publication until we have seen the revised manuscript and your response to the reviewers' comments. Your revised manuscript is also likely to be sent to reviewers for further evaluation.

Sincerely,

Daniel Bush

Academic Editor

PLOS Computational Biology

Thomas Serre

Section Editor

PLOS Computational Biology

Reviewer's Responses to Questions

**Comments to the Authors:**

Reviewer #1: Review of Bird, Cuntz and Jedlicka

This is a very nice paper, providing an extensive examination of information theory and pattern separation in the dentate gyrus. This is a complex study and is complicated by combining two complex and awkwardly discussed subjects in neural computation: information theory and pattern separation. However, I believe this paper does an exceptional job at using the former to help constrain the discussions of the latter, and as such, I think this is positioned to be an incredibly important study.

The main strength of this paper is its central premise – examinations in pattern separation in the dentate gyrus are poorly constructed because they fail to account for the information encoded by the neurons. This is, in large part, because the notion of pattern separation is often conflated with sparsity. One can get pattern separation in several different ways – one can get it by lowering firing rates in a randomly activated network – this was the premise of early DG studies like O’Reilly and McClelland [which the authors should probably cite] and Treves and Rolls; and one can get it from maintaining overall levels of activation yet actively selecting different neurons to be active, which is the premise of a lot [but not all] of studies modeling adult neurogenesis. Summarized briefly, “pattern separation” alone is an under-constrained concept and requires an accompanying notion of “information” to evaluate properly.

The paper discusses several formal measures of pattern separation and of information theory and relates them in a way which goes a long way towards achieving that goal. I have several comments that I think can make the paper stronger and more impactful.

• Provide a more direct and thorough historical context for the paper.

o This paper goes through a lot of references, particularly in recent years, that provide a strong grounding of the computational neuroscience relevant to this paper.

o The paper needs a bit more rigorous discussion of how in vivo physiology [and accompanying analyses and interpretations] has biased the perception of pattern separation in the dentate gyrus.

This is a very complicated field, made in part more complicated by the fact that the community which typically studies and quantifies in vivo activity in the dentate gyrus tends to not pay a lot of attention to neurogenesis [but see Rangel et al., Nature Communications 2014 and Rangel et al., Frontiers in Neurogenesis 2013 and Danielson et al., Neuron 2016] and uses rather old notions of pattern separation; while the computational literature in the field in recent decades has increasingly paid attention to the heterogeneity of the network offered by neurogenesis.

Fair or not, in vivo physiology drives the mainstream theoretical perspectives in hippocampus field. While the results described here are best studied and demonstrated in computational models [for which there is total control], the biggest impact of this approach will be taking these techniques into in vivo physiology data.

o More historical context around neurogenesis.

This paper does a great job of pointing to recent studies of neurogenesis, but I think it is fair to say the debate about information theory and pattern separation is fundamental to the neurogenesis field, going back to some of the earliest modeling and theory papers [most of which are not cited here]. The neurogenesis field has wrestled with the conflict that this paper addresses for longer than the authors may realize. In particular, Neuron published two papers back-to-back (a Point / Counterpoint) in 2011 [Aimone, Deng, Gage; Neuron 2011, Sahay, Wilson, Hen; Neuron 2011] that, in a sense, directly debate the question that is being addressed by this study. Summarized a decade later, the distinction is that the Sahay paper argued that new neurons role is to increase pattern separation through increasing sparsity by virtue of increasing feedback inhibition, whereas the Aimone paper argued that new neurons role is to increase information content [referred to as “memory resolution”] in the hippocampus.

• The paper combines single cell and population level pattern separation and information theory, which is awkward.

o As the authors note, pattern separation is a circuit concept, though can in certain circumstances be studied from a single neuron perspective. Likewise, information theory is best described for single neurons yet is challenging [but important] to scale up to circuits. I think this paper does a nice job of trying to bridge both scales, but it remains a bit awkward [to me] as I don’t have a great intuition of what the single neuron simulations (at moderate levels of complexity) mean. The information theoretic measures make sense, but I don’t really follow why one would even look at pattern separation here. That may be the point being argued [for single neurons, one should focus on information, not pattern separation], but I think the paper should be more explicit about this.

o While the authors cite several papers discussing pattern separation of individual neurons, there are other papers as well that have explored how differences of individual neurons can be involved in channel decorrelation [eg., Mishra and Narayanan, Hippocampus 2019, and other papers by those authors].

• The figures, overall, are informative and thorough but should be worked on.

o Somewhat obvious, but this paper has a lot of data presented in figures. I count something like 75 panels in just the main text alone (not counting the more populated supplemental figures). Adding to this, each figure has several sets of results plotted on them, and each figure has to be evaluated somewhat independently as the format of data and axes vary considerably across the paper.

o I would strongly suggest that the authors step back and ask for each figure what is the dominant message that is trying to be communicated, and ensure that *that* is what is best shown in the figure.

For instance, Figure 1 has a pretty fundamental point – if you filter data to sparsify it, you increase pattern separation but decrease information content. This result is rather consistent regardless of which filtering approach is used, which separation measure is used, and which information measure is used. The paper is not about filtering approaches, however, so it likely is sufficient to just show the random filter data [or whichever you choose] and only show the top left subpanel of each panel, and simply state that similar results are seen in the other filters. That will drive home the main takeaway message, and also allow the reader to better appreciate the differences between the separation and information metrics.

o I would also caution the authors from using figures that cannot be easily interpreted. For instance, the input and output panels in figure 3 are, I presume, intended to show how the neuron provides distinct outputs to different inputs. While the output figures (in particular) are attractive, they are too small to really interpret anything from.

• The network simulations are really small.

o Central to most of the theories about the dentate gyrus is its size relative to other regions. It has a lot of neurons. This is also somewhat central to the importance of neurogenesis, whereby new neurons represent a very small but correspondingly hyperactive set of neurons in the network.

o While a mouse DG is 300,000 neurons; this paper simulates 50 granule cells, with a half dozen or so interneurons. This is a really small number. Yes, there will be computational limitations due to the high fidelity of the neuron simulations; but there is no reason to only simulate populations of neurons at that level of fidelity. Further, it is 2022 and computing resources being what they are, the authors should be able to go beyond a few dozen neurons.

The main problem with networks of this size is the diversity of inputs. If multiple neurons in a network share a common set of inputs, they will end up with correlations that will propagate through all of these measures being considered. The degree of correlations of inputs is highly related to the scale of the inputs population, number of synapses, and number of neurons in the network.

A second problem with networks of this size is that adult neurogenesis is very connected to scaling – both in terms of the number of synapses arriving on immature neurons and the breadth of the sampling the new neurons take.

Stated differently, if the authors were to model 1000 GCs sampling 100 EC neurons (100,000 sampling 10,000) at a very low level of detail [and some of the papers the authors cite do just that], the dynamics will obviously be a lot less informative, but most of the pattern separation and information theoretic metrics should apply just the same. It wouldn’t surprise me if the authors see rather different behavior because the separation and information content is kind of baked into the sparse and dispersed projections.

Reviewer #2: See uploaded attachment

**Have the authors made all data and (if applicable) computational code underlying the findings in their manuscript fully available?**

Reviewer #1: Yes

Reviewer #2: Yes

PLOS authors have the option to publish the peer review history of their article (what does this mean?). If published, this will include your full peer review and any attached files.

Reviewer #1: No

Reviewer #2: No
---

## [Decision Letter · Decision Letter 1]

6 Jun 2023

Dear Dr Bird,

Thank you very much for submitting your manuscript "Robust and consistent measures of pattern separation based on information theory and demonstrated in the dentate gyrus" for consideration at PLOS Computational Biology. As with all papers reviewed by the journal, your manuscript was reviewed by members of the editorial board and by several independent reviewers. The reviewers appreciated the attention to an important topic. Based on the reviews, we are likely to accept this manuscript for publication, providing that you modify the manuscript according to the review recommendations.

In particular, the authors should make a few final minor changes to the wording of the manuscript, in order to clarify some of the key assumptions made and the relationship between this work and previous research, in line with the reviewers' comments. 

Sincerely,

Daniel Bush

Academic Editor

PLOS Computational Biology

Thomas Serre

Section Editor

PLOS Computational Biology

Reviewer's Responses to Questions

**Comments to the Authors:**

Reviewer #1: I think this paper is greatly improved from its original submission, which I enjoyed on my first reading as well. Thank you to the authors for addressing most of my concerns.

One minor technical point / question. In the new Figure 7 on the large scale simulation, the authors vary adult-born fraction from 0 to 1. In biological systems, the percentage of immature neurons is likely between 1% and 10% (depending on animal age and species); so the relevant range to examine is in between the first and second points on that graph. Since these experiments are at near mouse scales, a 1% rate is still meaningful. So it'd be interesting to run experiments where those low fractions of new neurons are examined. While this is a simple model, I'd expect that it is possible to see a non-linear "bump" of sorts with a very low rate of new neurons coming in.

Alternatively, the effect of neurogenesis may be different in different areas of the excitation - inhibition balance. For instance, does the location of the peak in Figure 7C or Figure 7D change if the proportion of new neurons changes?

Reviewer #2: The review was uploaded as an attachment

**Have the authors made all data and (if applicable) computational code underlying the findings in their manuscript fully available?**

Reviewer #1: Yes

Reviewer #2: Yes

PLOS authors have the option to publish the peer review history of their article (what does this mean?). If published, this will include your full peer review and any attached files.

Reviewer #1: **Yes: **James Bradley Aimone

Reviewer #2: **Yes: **Antoine D. Madar

Figure Files:

Data Requirements:

Reproducibility:

References:

---

## [Decision Letter · Decision Letter 2]

2 Oct 2023

Dear Dr Bird,

Thank you very much for submitting your manuscript "Robust and consistent measures of pattern separation based on information theory and demonstrated in the dentate gyrus" for consideration at PLOS Computational Biology. As with all papers reviewed by the journal, your manuscript was reviewed by members of the editorial board and by several independent reviewers. The reviewers appreciated the attention to an important topic. Based on the reviews, we are likely to accept this manuscript for publication, providing that you modify the manuscript according to the review recommendations. Specifically, some of the language in the manuscript still needs to be edited for accuracy, and some of the finer details of the Methods included, before we can move forward.

Sincerely,

Daniel Bush

Academic Editor

PLOS Computational Biology

Thomas Serre

Section Editor

PLOS Computational Biology

In particular, some of the language in the manuscript still needs to be edited for accuracy, and some of the finer details of the Methods included, before this paper can be accepted for publication.

Reviewer's Responses to Questions

**Comments to the Authors:**

Reviewer #2: I have two remaining comments with which the authors have not really engaged:

1) Upsilon_M as a measure of Pattern Separation (Line 256 and 291 in the marked-up manuscript):

Despite the authors’ response to my comments #2 and #3, I still think it is erroneous to call Upsilon_M a measure of pattern separation, as it does not directly compare measures of pattern similarity. Even if it necessarily correlates with pattern separation (which is not demonstrated in the paper), that's not what it measures. It measures efficient coding (sparse and informative). The argument that sparse information transmission is equivalent to pattern separation in a high dimensional space is a very interesting idea, but this argument is not explicitly laid out in the manuscript (only partially in the rebuttal). A citation of Ganguli and Sompolinski 2012 (which does not directly talk about pattern separation) is not enough: if the authors want to claim (and convince) that their metric is the better way to think about pattern separation, they need to make a rigorous demonstration that there is an equivalence or a correlation. And, at the very least, instead of calling it a “measure of pattern separation”, I think it would be safer to call Upsilon_M a “proxy” for pattern separation. The field of pattern separation is already ridden with confusing terminology applied differently in different context, best not to add to the confusion.

2) Reporting the codes and binsizes selected by the optimization procedure (response to comment 1d in previous review + see line 284 of the marked-up manuscript):

I understand that this information is not easy to summarize. Still important to report here, to provide the reader with a full understanding of the data presented in the manuscript. The full info (not summarized) could be made available as a supplementary table. For a more palatable summary, one could imagine a plot of the distribution of binsizes over all analyses (one plot for inputs, one for outputs), to see if they vary widely or if we always end up in the same ballpark. And for some of the most important analyses (e.g. Fig 3 and 6), the authors could at least report the neural codes and binsizes that maximized the used metric. These are just ideas, there might be better ones, the goal being to make the optimization procedure less of a black-box to the reader. There is some interesting info there.

Minor comment:

Line 602-4 (in marked-up manuscript), the authors added more comparison on methods used to measure pattern separation in past studies (use of a ratio vs a difference between input and output similarities). I think some terms used are too vague (it is hard to relate to the problem of ratio vs difference of pattern separation) and sometimes inaccurate: 1) not sure what is meant by “overlap” (in reference to the Stark review). Overlap is a measure of similarity, not of pattern separation, isn’t it?, 2) “integrated over a range”, in reference to the Guzman study: not clear a range of what? How does that relate to the ratio vs difference approaches? 3) As far as I understand, the Yim and Wolfart 2014 study does not use ratios but plots input vs output similarities, like in the Madar studies. Most studies or analyses actually do not make a decision in this ratio vs difference debate, they just plot the similarities, allowing a direct visual comparison.

**Have the authors made all data and (if applicable) computational code underlying the findings in their manuscript fully available?**

Reviewer #2: Yes

PLOS authors have the option to publish the peer review history of their article (what does this mean?). If published, this will include your full peer review and any attached files.

Reviewer #2: **Yes: **Antoine Madar

Figure Files:

Data Requirements:

Reproducibility:

References:

---

## [Editor Report · Decision Letter 3]

13 Dec 2023

Dear Dr Bird,

We are pleased to inform you that your manuscript 'Robust and consistent measures of pattern separation based on information theory and demonstrated in the dentate gyrus' has been provisionally accepted for publication in PLOS Computational Biology.

Best regards,

Daniel Bush

Academic Editor

PLOS Computational Biology

Thomas Serre

Section Editor

PLOS Computational Biology

---

## [Editor Report · Acceptance letter]

9 Jan 2024

PCOMPBIOL-D-22-01609R3 

Robust and consistent measures of pattern separation based on information theory and demonstrated in the dentate gyrus

Dear Dr Bird,

I am pleased to inform you that your manuscript has been formally accepted for publication in PLOS Computational Biology. Your manuscript is now with our production department and you will be notified of the publication date in due course.

With kind regards,

Livia Horvath
